# Magnetoreception in a freshwater ciliate arises from endosymbiosis

Romain Bolzoni[1,2,7], Caroline L. Monteil [1,7] ✉, Béatrice Alonso[1], Marine Bergot[1], Daniel M. Chevrier [1], Christian Godon[1], Nicolas Menguy[2], Stephanie Fouteau [3], Violette Da Cunha[3], Fériel Skouri-Panet[2], Eva Pereiro[4], Arnaud Duverger [2], David Vallenet [3], Corinne Cruaud[5], Fernanda Abreu [6], Karim Benzerara [2] & Christopher T. Lefevre [1] ✉

Magnetoreception is a remarkable ability found across a diverse range of organisms, including bacteria, birds, fish, insects, and mammals, enabling them to detect and harness the Earth's geomagnetic field. Recently, the recruitment of biomineralizing ectosymbionts by euglenozoans was evidenced as an ecological strategy for microeukaryotes to acquire this sense. Here, we report a case of magnetosymbiosis involving a ciliate and four populations of endosymbiotic bacteria experiencing genome reduction. Among these bacteria, one group of sulphate-reducing *Desulfovibrionales* was found to biomineralize bundles of bullet-shaped magnetite crystals. The ciliate's magnetotaxis mirrors that of free-living magnetotactic bacteria and euglenozoans, enabling efficient navigation in chemically stratified aquatic environments. However, in this case, magnetotaxis arises from an endosymbiotic interaction. Using a combination of optical-, confocal-, electron- and X-ray-based microscopy techniques, together with genomic analyses, these findings demonstrate that magnetosymbiosis can emerge in unicellular eukaryotic lineages through endosymbiotic integration, expanding our understanding of such interactions in aquatic ecosystems. More broadly, this work contributes to the ongoing debate on the origins of magnetoreception in eukaryotes.

Species biology has long been studied considering organisms as independent, competing functional units. Today, however, it is clear that the evolution and functioning of living organisms can only be fully understood by integrating the role of additional levels of biological complexity and symbiotic interactions. Symbiosis is now recognized as one of the major forces that has shaped biodiversity[1,2]. It refers to the long-term coexistence of at least two species, where at least one depends on the other, regardless of whether the interaction is beneficial or detrimental to their fitness[3]. This interaction may therefore be based on conflict or cooperation, which can often be mediated by the same molecular mechanisms[4,5].

Symbioses are multi-scale systems; partners and associations may have different degrees of biological complexity, and a network of intimate interactions may be established between several partners on a

[1]Aix-Marseille Université, CEA, CNRS, UMR7265, BIAM Biosciences and Biotechnologies Institute of Aix-Marseille, CEA Cadarache, Saint-Paul-lez-Durance, France. [2]UMR CNRS 7590, IMPMC Institut de Minéralogie, de Physique des Matériaux et de Cosmochimie, MNHN, IRD, Sorbonne Université, Paris, France. [3]LABGeM, Génomique Métabolique, Genoscope, Institut François Jacob, CEA, CNRS, Université d'Évry, Université Paris-Saclay, Evry, France. [4]ALBA Synchrotron Light Source, Cerdanyola del Vallés, Barcelona, Spain. [5]Genoscope, Institut de biologie François Jacob, CEA, Université Paris-Saclay, Evry, France. [6]Instituto de microbiologia Paulo de Goés, Universidade Fedral do Rio de Janeiro, Rio de Janeiro, Brazil. [7]These authors contributed equally: Romain Bolzoni, Caroline L. Monteil. ✉e-mail: caroline.monteil@cea.fr; christopher.lefevre@cea.fr

permanent or transient basis, either as obligatory or facultative relationships. Symbionts may be simply attached to host cells (ectosymbiosis) or become integrated within the host cell (endosymbiosis). In their entirety, each host-symbiont consortium forms a supra-organism, i.e., an ecologically cohesive and functional unit also defined as a holobiont[6]. The level of physical integration and dependence of symbionts can even challenge the definition of the term "organism", as some of them have progressively evolved into organelles over time[2].

Today, the role of symbiosis and cooperation in the functioning and evolution of the microbial world is widely acknowledged, reaffirming the pioneering ideas of Lynn Margulis[6,7]. These mechanisms have gradually become central to evolutionary models of eukaryogenesis and organelle acquisition[3,8]. According to these scenarios, multiple independent and/or successive endosymbiosis events involving *Archaea*, *Pseudomonadota* (previously referred to as *Proteobacteria*), *Cyanobacteria*, and/or ancestral eukaryotic lineages led to the diversification of modern eukaryotic lineages[9]. These interactions led to the emergence of plastids, mitochondria, and potentially also the nucleus. Many organelles, as we know them today, have thus derived from the invagination, slavery, or synergistic fusion of endosymbionts, granting several hosts novel functions, such as photosynthesis, nitrogen fixation, aerobic or anaerobic respiration.

Research in this field currently suffers from three weaknesses. First, much more attention has been paid to microbial symbioses involving at least one eukaryotic model macroorganism host[10]. As a result, most of our knowledge is based on microbiota associated with model macroorganisms (e.g., humans, corals, insects, and plants)[11–14]. Second, the role of conflict in host-pathogen and host-parasite systems has been much better characterized than that of cooperation in mutualistic or commensal systems[15]. Most of the current knowledge is therefore based on the mechanisms and evolution associated with a single type of interaction. Finally, several technological hurdles have long limited the observation of microbial holobionts, the identification of partners, and the visualization of their interaction. By overlooking holobionts involving eukaryotic-prokaryotic and prokaryotic-prokaryotic microorganisms, especially those based on cooperation and mutualism, we have neglected an entire dimension of molecular, biochemical, physiological, and evolutionary mechanisms that govern species interactions across countless biological systems.

The establishment of symbiosis has often enabled hosts to "jump" into new ecological niches by conferring new capabilities. Interactions and benefits are often based on nutrition, detoxification, and energy production through the exchange of metabolites, carbon or nitrogen sources, and electron acceptors/donors[16]. Such interactions gave rise to trophic dependency, also known as syntrophy. Photosynthesis is the most emblematic case[17,18]: a freshwater endosymbiotic cyanobacterium led to the emergence of the first photosynthetic eukaryote and its descendant lineages[19]. Additionally, some photosynthetic algae granted the ability to sense light and phototaxis[20] to some warnowiid dinoflagellates. Another well-documented case is that of the anaerobic flagellates in the digestive tract of termites, themselves in symbiosis with their insect host[21], some of which supply nitrogenous elements to endosymbiotic bacteria of the order *Bacteroidales*[22]. Several taxa also evolved hydrogen-based syntrophy to adapt to anaerobic conditions[23,24]. For example, some ciliates and amoebae-like protists produce molecular hydrogen from mitochondria-related organelles, which their symbionts use to reduce sulfate and produce energy[25,26]. Some predatory flagellated or ciliated protists can also use symbionts as a primary source of nutrients. For example, ciliates of the genus *Kentrophoros*, can cultivate *Gammaproteobacteria* and phagocyte them as a source of food[27]. Beyond syntrophic interactions, symbiosis confers additional advantages such as protection and motility. Certain ectosymbioses provide protection against competitors or predators, as seen in *Euplotidium* spp., which partner with *Verrucomicrobia*, expulsing harpoon-like structures when triggered[28]. Other microeukaryotes are propelled by the movement of ectosymbiotic spirochetes in *Mixotricha paradoxa*[29].

New associations continue to be discovered. In 2019, we described a novel ectosymbiosis between an euglenozoa and a bacterium affiliated with *Symbiontida* and *Desulfobacterales*, respectively, isolated from anoxic marine sediments of Carry-le-Rouet, France[30]. This relationship relies on classical hydrogen-based syntrophy involving sulfate-reduction, acetogenesis, and hydrogen transfer from mitochondria-related organelles in the host. While similar syntrophic cases had been described in other protists inhabiting the same niche before[3,26,31], this holobiont introduced a new collective behavior based on magnetotaxis. Collective magnetotaxis involves the production and alignment of prokaryotic organelles known as magnetosomes, into chains along the symbiont and host motility axis, coupled with host-driven chemo-aerotaxis[30,32]. Together, the eukaryotic host and bacterial symbionts form a magnetosymbiosis, reproducing the same magnetoreception-driven motility as for free-living magnetotactic bacteria[32]. This behavior is an adaptation to chemically stratified environments, whereby organisms move efficiently towards their optimal redox niche, combining guidance by the Earth's magnetic field lines and chemotaxis. Prior to this discovery, other magnetotactic protists had been observed[33,34], but their magnetotaxis was transient only, relying on the grazing of magnetotactic bacteria accumulated in acidic food vacuoles. Meanwhile, some freshwater protists were hypothesized to exhibit magnetotaxis, biomineralizing magnetosomes within their cytoplasm[35]. However, the rarity of their observations in sediments from the Uruai and Ubatiba rivers (Brazil) has hampered comprehensive genetic and ultrastructural analyses necessary to confirm these conclusions. Altogether, these recent observations have prompted the search for additional magnetosymbioses in aquatic habitats that could be more widespread than currently known.

In this work, through multiple exploratory sampling campaigns, we identified a specific site along the Dordogne River (France) where magnetotactic ciliates thrive, morphologically resembling those documented by Leao et al.[35]. A comprehensive characterization was carried out using culture-independent molecular approaches, advanced microscopy, and synchrotron analyses to elucidate the origin of their magnetotactic behavior. We demonstrated that these unicellular eukaryotes are, in fact, multi-partner holobionts composed of a ciliate (*Ciliophorea*, *Prostomatea*) and four populations of endosymbiotic bacteria, one of which, belonging to the *Desulfovibrionales*, biomineralizes a bundle of bullet-shaped magnetosomes. We concluded that magnetosymbiosis has independently emerged in another protist phylum through endosymbiosis.

## Results

### Observation of anaerobic magnetotactic protists in freshwater sediments

Standard magnetic enrichment was performed on sediments from various sampling sites in France to target magnetically responsive microeukaryotes. Since the first identified magnetosymbiosis was south-seeking in the Northern Hemisphere, which is uncommon[30], both south-seeking and north-seeking microorganisms were sorted and observed. Magnetotactic bacteria (MTB) were found in nearly all sampled locations. In addition to diverse MTB, a singular north-seeking, magnetically responsive organism was initially observed at a few sites in the Dordogne River near Beaulieu-sur-Dordogne (44°58′55.5″N, 1°50′14.5″E) (Fig. 1a, b; Supplementary Videos 1 and 2). The cells were rod-shaped, measuring $26.7 \pm 2.3\,\mu m$ in length and $10.9 \pm 1.3\,\mu m$ in width ($n = 19$), suggesting they were likely protists. Their strong alignment and responsiveness to the reversion of the magnetic field indicated active magnetotaxis rather than passive magnetism due to the grazing on MTB[34]. The cells had a helical motion facilitated by numerous short flagella surrounding them, visible under the light microscope (Fig. 1a–c and Supplementary Video 2). When reaching the

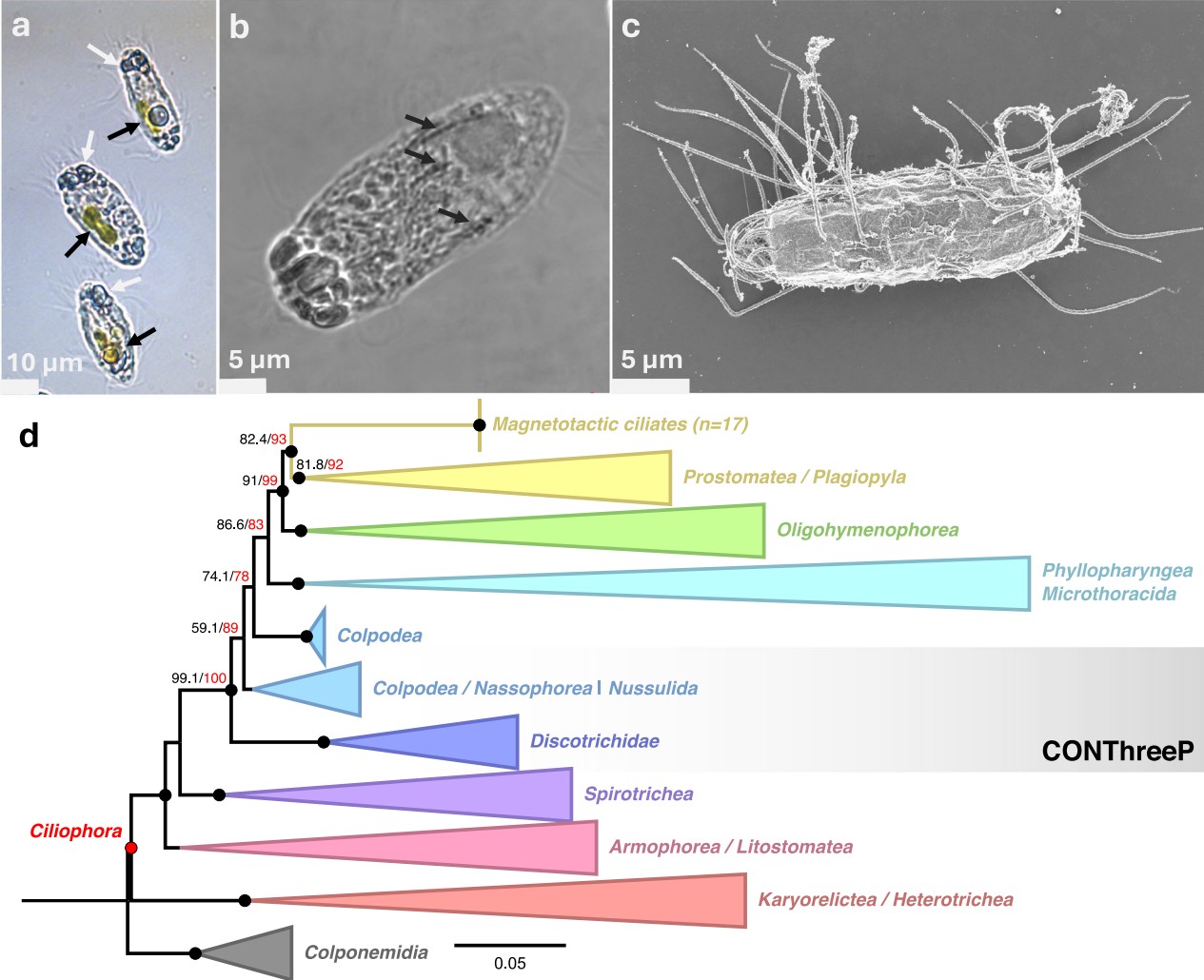

**Fig. 1 | Images and identification of the magnetotactic protist from the Dordogne River, France. a** light microscopy images of fixed cells between slide and coverslip showing the numerous flagella or cilia surrounding the protists, their contrasted frontal part (white arrows), and the presence of greenish to brownish inclusions within their cytoplasm (black arrows). **b** Transmitted CLSM image of a magnetotactic protist showing contrasted granules in the anterior part and dark lines aligned in the anteroposterior axis (black arrows). **c** Scanning electron microscopy image in the secondary electron detection mode. **d** Maximum-Likelihood tree based on the SSU 18S rRNA gene sequences showing the evolutionary relationships of the magnetic ciliates with other *Ciliophora* classes. A full and uncollapsed tree is given in the Supplementary Data 1 and 2 with the full PR2 accession numbers and taxonomy. The scale represents the substitution rate. A SH-aLRT branch test and ultrafast bootstrap approximation estimated from 1000 replicates were conducted to provide branch support values in red and black, respectively. Black circles highlight branches supported with a aLRT value > 80 and a proportional bootstrap support value > 90. The magnetotactic ciliates 18S rRNA sequences were submitted to the NCBI Genbank database under the accession numbers of OR342250.1 to OR342267.1.

edge of the droplet, the cells moved back and forth while the magnetic field remained unchanged (Supplementary Videos 1 and 2). This so-called "ping-pong motion" was previously described in multicellular magnetotactic prokaryotes[36]. Magnetic sorting at different sediment depths along the oxygen gradient showed that these microorganisms lived below the oxic-anoxic transition zone. The cells appeared sensitive to mechanical force as (1) the number of magnetotactic protists harvested using the magnet after magnetic concentration was higher when the magnetic enrichment was performed with a weak stirring bar magnet (~0.4 mT) for a maximum of 2 h, and (2), the cells survived for up to 10 min under a hanging drop, while they rounded and exploded after less than 1 min when transferred between a slide and a coverslip (Supplementary Video 3).

These protists were particularly abundant in river areas rich in humic acids and leaf litter, with minimal water current exposure. Similar magnetotactic protists exhibiting a comparable magnetotactic behavior were found in other freshwater habitats in France, including an unnamed spring (47°45'13.7"N, 3°28'54.1"W) (Supplementary Video 4), lake Lannénec (47°44,29.9", 3°28'58.7"W) in Britany, Lake Aydat in Auvergne (45°39'54.1"N, 2°58'53.1"E) and the Cère River (44°55'1.3"N, 1°50'18.8"E), a few kilometers upstream from its confluence with the Dordogne River. For practical reasons, this study focuses on the description of populations observed in the Dordogne River, as it was the most easily accessible for sampling.

## The magnetotactic protist belongs to an undescribed group of ciliates within Prostomatea (Alveolata)

The morphology of the magnetotactic protist initially observed under light microscopy was further defined using scanning electron microscopy (SEM). This revealed a ciliate with a simple ovoid to cylindrical body, featuring a wide oral bulge and opening in apical position at the anterior pole and surrounded by cilia with a membranelle-like arrangement (Fig. 1c and Supplementary Fig. 1a−e). This organization consisted of twelve sets of 15-µm-long microtubules, arranged in a twisted pattern resembling a cyrtos (Supplementary Fig. 1). The external part of this cyrtos-like structure is covered by the cilia of the

membranelle and presents similarities with a tooth-like capitulum (Supplementary Fig. 1). The cytostome opening measures about 0.5 μm, likely restricting predation to small bacterial prey.

Each of the apical cilia extended to the cell body as microtubules reaching the extremity of the posterior pole (Fig. 1c; Supplementary Fig. 1b and 2a). Unlike previously described marine magnetotactic *Euglenozoa*[30], no epibionts were observed; instead, the entire ciliate body was covered by a mucus-like envelope. Beneath this envelope, the cell membrane was homogeneously perforated with pores (Supplementary Fig. 2a, b). Cilia were aligned with the cell's anterior-posterior axis along each longitudinal microtubule (Fig. 1c, Supplementary Fig. 1a, b). Transverse thin sections of several cilia revealed the canonical microtubule arrangement with 9 doublet microtubules in a circle and two central microtubules (Supplementary Fig. 2c) as usually described for flagella and cilia in eukaryotes. Some rare observations indicated that the magnetotactic protists reproduce by binary fission with the cleavage furrow forming at the equator (Supplementary Fig. 3a), while other observations suggested that they also perform sexual reproduction (Supplementary Fig. 3b).

Molecular typing of several single magnetotactic ciliates sorted from four different sites where they were observed yielded 17 nearly identical sequences of 18S rRNA gene (>99.9% similarity). They form a monophyletic clade together with *Prostomatea* and *Plagiopylea* within the CONThreeP superphylum (Fig. 1d and Supplementary Data 1). Pairwise BLASTN alignments were performed with sequences of these two genetically closest groups retrieved from the PR[2] database (https://app.pr2-database.org). Although average pairwise identity values between members of *Prostomatea* and *Plagiopylea* were approximately 90% and 85%, respectively, they were on average 86% and 82% identical with those of the magnetotactic protists, respectively. These genetic distances, along with their basal phylogenetic position relative to known families, placed these organisms *incertae sedis* in the paraphyletic class *Prostomatea*. Alternatively, the data could support their classification as a new undescribed class within CONThreeP. Pending a more formal taxonomic description in the future, we named the ciliate HBD-1 (HoloBiont Dordogne-1).

### The ciliate harbors intracellular magnetite-bearing vesicles, accessory photosynthetic organisms, and quartz

Three distinct types of inclusions were observed within magnetotactic protists under the light microscope: (i) one to three greenish to brownish oblong structures aligned along the antero-posterior axis and filling about 1/4 of the cell body (Fig. 1a), (ii) contrasted granules surrounding the cytostome at the anterior pole (Fig. 1a, b; Supplementary Videos 2 and 5) and (iii), five to six thin, dark and contrasted lines concentrated in the anterior two-thirds of the cells (Fig. 1b and Supplementary Video 5). Several minutes of exposure to air led to the disintegration of the magnetotactic protists, releasing their intracellular contents. The first type of structure was identified as diatom-like skeletons that, based on the shape of the frustules, could represent different species within a single cell (Fig. 1a; Fig. 2 and Supplementary Video 6). Elemental mapping conducted on a single protist using synchrotron-based nanobeam-scanning X-ray fluorescence (nano-XRF) confirmed the silica-based nature of these skeletons (Fig. 3a). Some chloroplasts and thylakoids appeared preserved, contributing to the greenish-brownish color inside the magnetotactic protists (Figs. 1a and 2a, b). Although diatoms were the predominant intracellular photosynthetic microorganisms, TEM images of thin-sectioned cells showed the presence of other microalgae, with ultrastructural features resembling *Chlorophyta* species (Supplementary Fig. 4). We did not observe any protist with dismantled diatoms, but some were observed without intracellular diatoms.

Nano-XRF also detected silica in the second structure, i.e., within the contrasted polar inclusions localized at the anterior pole surrounding the oral opening. Diffraction analyses indicated that these inclusions are crystalline, corresponding to quartz (Fig. 3b and c). Their morphology, size, and number vary across cells, but they are consistently located at the anterior pole. They are easily fractured by the diamond knife during the thin section preparation (Supplementary Fig. 1f, g), similar to the diatom skeleton.

TEM and Nano-XRF analyses of the third structure visible under optical microscopy as thin, dark, and contrasted lines, revealed that they consist of particles similar in size and shape to bullet-shaped magnetosomes found in magnetotactic bacteria[37] (Fig. 3a, d, e and Supplementary Video 5). They form bundles of three to five chains filling almost entirely 5-6 rod-shaped vesicles that resemble cells, enclosed by a lipid-like bilayer (Fig. 3d, e). Unlike magnetosomes in grazers of magnetotactic bacteria[34], all these particles are intact, and none of the cell-like inclusions are located within digestive vacuoles. High resolution TEM (HRTEM) coupled with elemental analysis confirmed these particles are rich in iron and made of magnetite (Fig. 3a and Supplementary Fig. 5). Each vesicle contains about $131 \pm 26$ particles ($n = 13$), which measure $143.6 \pm 40.9$ nm in length and $55.2 \pm 8.8$ nm in width ($n = 629$), and are surrounded by a thin membrane similar to the membrane generally observed surrounding bullet-shaped magnetites[38] (Supplementary Fig. 5d). Some particles are as large as $282.8 \times 62.1$ nm (Supplementary Fig. 5a). TEM images of thin-sectioned cells showed these vesicles surrounded by many other similar structures lacking magnetite particles, distributed throughout the protist (Fig. 4a, b).

### The ciliate cytoplasm is filled with intracellular bacteria

We tested the hypothesis that rod-shaped vesicles filling the protist cytoplasm, including those with magnetic particles, were in fact bacteria. For this purpose, we sorted single magnetotactic protists and applied fluorescent in situ hybridization (FISH) using a fluorescent probe targeting *Bacteria*. Confocal laser scanning microscopy (CLSM), combined with previous TEM analyses of thin-sectioned cells, validated our hypothesis and revealed the presence of 50–100 bacteria within each protist, along with diatoms (Fig. 4). All bacteria appeared to be fully enclosed within their host, with no detectable connection to the plasma membrane or the extracellular medium, and were therefore considered endosymbionts. The bacteria were rod-shaped and, when observed outside the disaggregated protist, flagella were never observed. The intracellular organization of the protist conserved in near-native form was imaged with cryo-soft X-ray tomography (cryo-SXT) (Supplementary Video 7, diatom and four magnetosome-bearing bacteria are labeled). Bacteria appear to be embedded in an organic compound surrounded by a membrane-like envelope, but none were located within the digestive vacuoles of the host. The bacterial cells measured approximately 5 μm in length and 0.5 μm in width. Most of them exhibited a uniform electron contrast, except for 5–10 cells that contained electron-dense particles (Fig. 4). These observations confirm that the magnetotactic protist forms a microbial holobiont with endosymbiotic bacterial species. By validating the cellular nature of these structures, we concluded that particles were, in fact, magnetosomes sensus lato formed by some of the endosymbionts. Additionally, structures resembling mitochondria-related organelles, i.e., hydrogenosomes[39], were observed enveloping some endosymbiotic bacteria, including those with magnetosomes (Fig. 4b). Similar observations were made in protists isolated from Lake Aydat in Auvergne (Supplementary Fig. 6a), the unnamed spring in Britany (Supplementary Fig. 6b–d), and Lake Lannénec, Britany (Supplementary Fig. 6e, f).

### Intracellular bacteria belong to four taxonomic groups, some of which are known for their symbiotic lifestyle

The identity of the endosymbionts was investigated through pyrosequencing of cloned 16S rRNA gene sequences of sorted magnetotactic holobionts (MHB) collected from the Dordogne River and three other sites where these magnetotactic ciliates were found

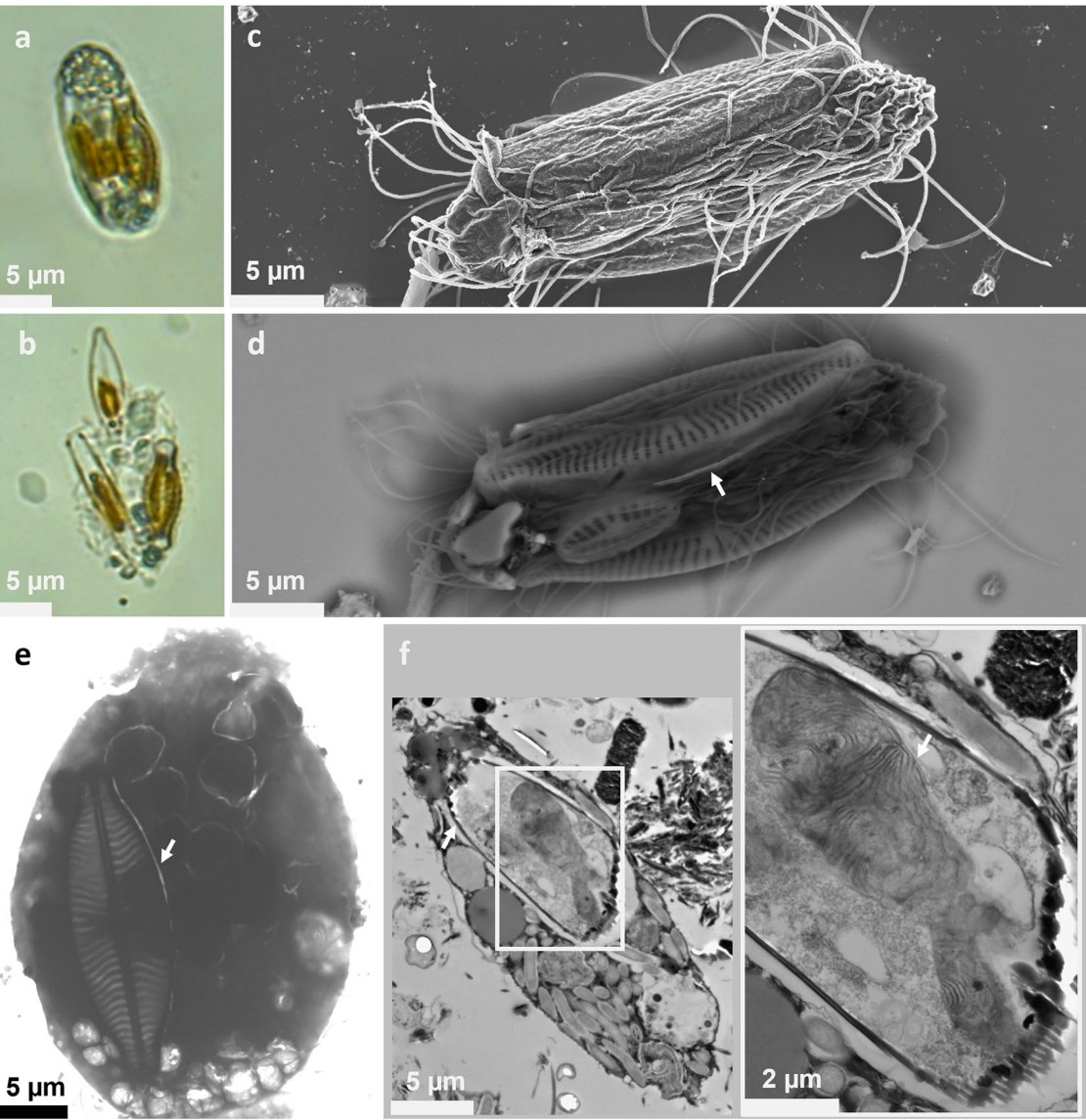

**Fig. 2 | Microscopy images showing the presence of diatoms inside the magnetotactic protists.** Light microscopy images of the same magnetotactic protist before (**a**) and after (**b**) it breaks apart, showing the presence of three diatoms of different shapes inside the protist. These images are from the Supplementary Video 6. **c**, **d** Images of a single magnetotactic protist observed using a scanning electron microscope operating at 3 kV using the InLens mode−secondary electron (**c**) and 10 kV using the AsB mode−backscattered electron (**d**) showing the presence of three diatoms inside the protist. Note that the presence of diatoms inside the protist can be inferred even in the low-voltage image due to the deformation of the external membrane of the protist. The white arrow shows a bright (i.e., electron-dense) line parallel to the long axis of the protist. **e** Transmission electron microscopy (TEM) image of a whole magnetotactic protist deposited by micro-manipulation on the TEM grid showing the presence of a diatom inside the protist (white arrow). **f** TEM images of a longitudinal thin-sectioned protist showing the presence of a diatom inside the protist. The higher magnification image, corresponding to the white frame in the panel on the left, shows the conservation of the intracellular content of the diatom, including thylakoids (white arrow).

(Supplementary Fig. 7a–d). Whole genome sequencing of one representative hologenome HBD1-SC9 was also carried out after genome amplification. Three high-quality bacterial genomes (i.e., >90% complete and <5 % redundancy according to checkM values on GTDB release 220) and one medium-quality genome (i.e., 86.17% complete and 0.77% redundant) were obtained using both long- and short-read technologies (Supplementary Data 4). However, only a few eukaryotic reads were obtained after the hologenome amplification−approximately 31 million, representing 15.3 % of the total reads. They mapped to 2 209 contigs representing a total assembly length of 19.95 Mbp. According to the Protista_83 single-copy core gene (SCG) collection in anvi'o[40], the assembly corresponds to a highly incomplete genome (<6.0% completeness) with substantial redundancy (>9.6%). Only a single 18S rRNA gene sequence associated with the ciliate was

detected. A 16S rRNA gene sequence was assembled with each of the four bacterial bins of the single amplified hologenome. These four 16S rRNA gene sequences were identical to those obtained by multiple amplicon sequencing of various MHB samples.

Molecular phylogenetics revealed the identities of the four endosymbiotic populations detected in the MHB (Fig. 5a–d). The first two endosymbiotic populations belong to two groups of sulfate-reducing bacteria within the *Desulfobacterota* phylum. One population, represented by the DsD-1 genome, belongs to a diverse family well described in the literature: the *Desulfovibrionaceae* (Fig. 5a), which includes several freshwater MTB, such as the reference magnetotactic bacterium *Solidesulfovibrio magneticus* RS-1[41]. GTDB-Tk v2.1.1[42] assigned DsD-1 to an undescribed genus forming a monophyletic group with both *Oceanidesulfovibrio* and *Megalodesulfovibrio* genera,

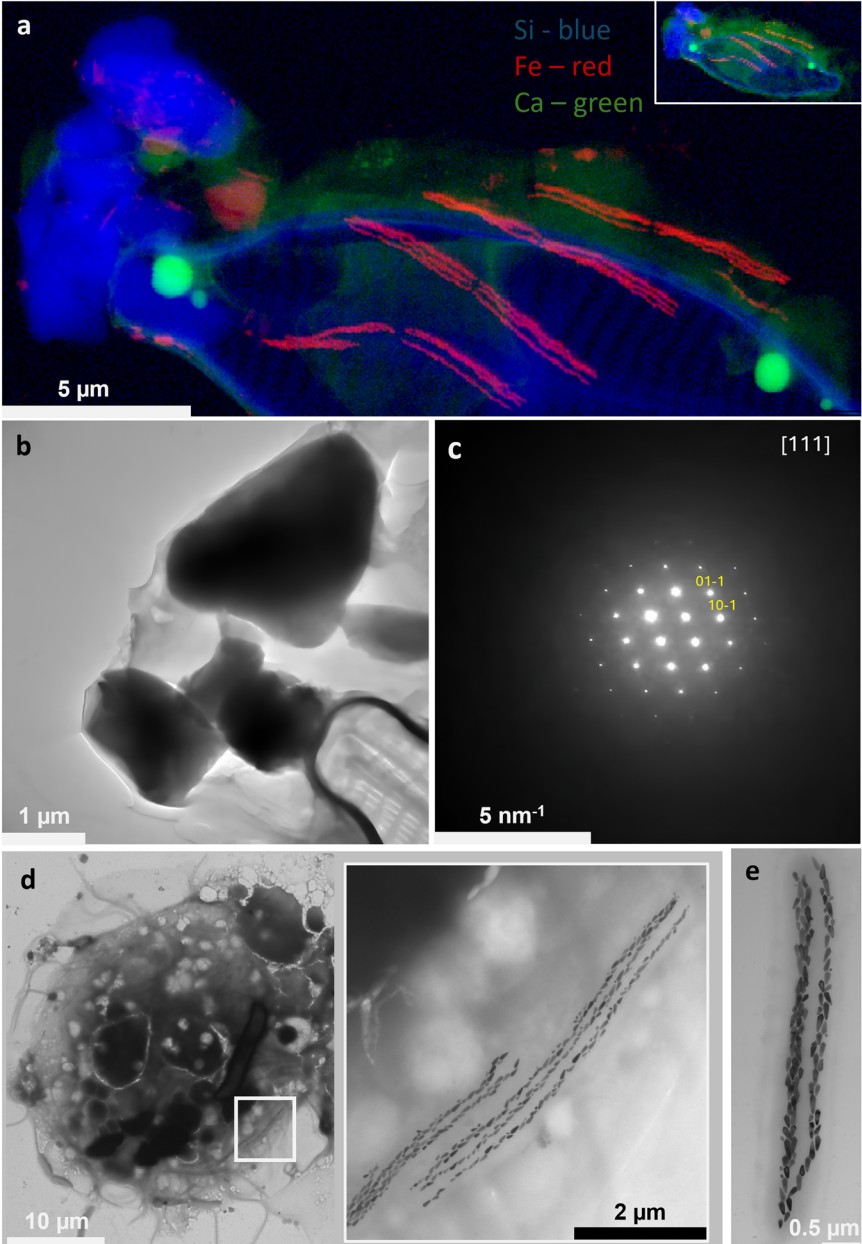

**Fig. 3 | Analyses of the intracellular structures present in the magnetotactic protist. a** Synchrotron-based nanobeam-scanning X-ray fluorescence showing the presence of Si in the frontal part of the protist, Si in the diatom skeleton and Fe in the first third of the anterior pole of the cells. The inset (top right image) is the image of the whole cell corresponding to a low-resolution map. **b** Transmission electron microscope image of the frontal part of a magnetotactic protist showing electron-dense particles. Note the presence of part of a diatom (bottom right). **c** Diffraction analysis of a Si-rich particle on the frontal part of the magnetotactic protist of image b, showing a crystalline structure. The diffraction is indexed with the quartz structure ([111] shows the direction of observation). **d** TEM images of a whole magnetotactic protist showing in the white frame the presence of dark and contrasted lines similar to bullet-shaped magnetosomes found in magnetotactic bacteria. **e** TEM image of the dark chains composed of magnetosome-like structures outside the protist reveals that these particles are enclosed within a membrane-bound vesicle resembling a bacterium.

composed of free-living bacteria. The second population, represented by the DcD-1 genome, is affiliated with a poorly known family: the *Desulfarculaceae* (Fig. 5b), for which few genomes and 16S rRNA gene sequences are available. This family has long been a separate and distant lineage within the former *Deltaproteobacteria* class and is not known for being a group of intracellular symbionts[43–45]. GTDB-Tk v2.1.1[42] assigned DcD-1 to an undescribed genus g_JACRDC01 with no representative cultivated strain.

The third symbiotic species belongs to the *Ca.* Midichloriaceae family: a diverse but understudied lineage within the important order of intracellular bacteria *Rickettsiales* of the *Alphaproteobacteria* class[46].

GTDB-Tk v2.1.1[42] assigned the HBD1-SC9 endosymbionts (represented by the MD-1 genome), to an undescribed genus clustering with endosymbiotic species of another family of ciliates (*Oligohymenophorea*) isolated from a wastewater treatment plant in Rio de Janeiro (Brazil)[47] (Fig. 5c). Although the phylogenetic tree based on the 16S rRNA gene sequences is weakly supported, these endosymbionts appear to also cluster with the genus *Ca.* Euplotella, endosymbionts of ciliates from the same family[48].

The fourth population affiliates with the family *Coxiellaceae* (*Gammaproteobacteria* class; *Legionellales* order) (Fig. 5d). This group is best known for being represented solely by intracellular bacteria

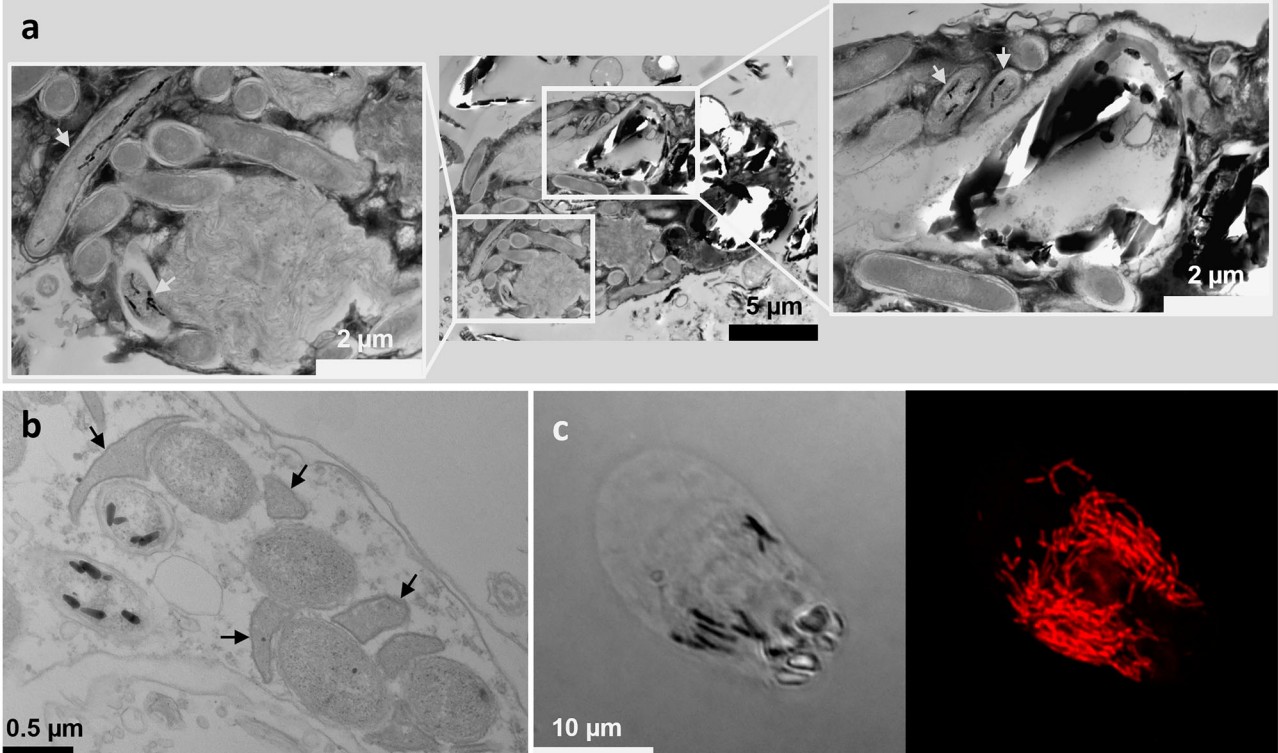

**Fig. 4 | Micrographs showing the presence of numerous bacteria inside the magnetotactic protist. a** Transmission electron microscopy (TEM) images of a thin-sectioned magnetotactic protist reveal the presence of numerous endosymbiotic bacteria, four of which contain magnetosomes (white arrows indicated in the left and right images). **b** TEM image showing the presence of hydrogenosomes-like structures (black arrows) surrounding the endosymbiotic bacteria with and without magnetosomes. **c** Fluorescence in situ hybridization (FISH) of a magnetotactic protist (transmitted light on the left panel) showing the presence of numerous rod-shaped bacteria whose 16S rRNA gene sequence hybridized with an eubacterial probe (red fluorescence on the panel on the right).

infecting microeukaryotes and opportunistic pathogens in humans[49]. A growing number of environmental surveys highlight the underestimated diversity of this group and their eukaryotic hosts, including arthropods and microeukaryotes such as amoebae[49]. This classification within *Legionellales* and this family specifically, might be revised in the future, as GTDB currently splits them into different orders[50]. Indeed, GTDB-Tk v2.1.1[42] assigned the fourth population of HBD1-SC9 endosymbionts (represented by the genome GD-1) to an undescribed order o_UBA9339 clustering with the order *Coxiellales*[51] represented by the obligate intracellular pathogen *Coxiella burnetii*[49]. The few small-sized genomes (<1.5 Mbp) within the same cluster were assembled from metagenomes of diverse aquatic sediments across the world, and none have been linked to cultivated organisms (Fig. 5d). Few close 16S rRNA gene sequences of uncultured bacteria associated with various freshwater and saline habitats could be retrieved from public databases (Supplementary Fig. 7d).

**Endosymbionts, including magnetosome-forming Desulfovibrionales, exhibit a specific pattern of cellular organization**
FISH assays, using specific probes designed from the 16S rRNA gene sequence of each of the four bacterial populations, confirmed the taxonomic affiliation of the different endosymbiotic bacteria, and further identified their spatial distribution and relative abundance (Fig. 6a and Supplementary Videos 8, 9). Multiple FISH and TEM analyses carried out with different MHB from the Dordogne River revealed a unique organizational pattern, where each symbiont is systematically observed in the same position within the host cytoplasm. Bacteria producing magnetosome bundles were stretched across the whole length of the host as shown in Supplementary Videos 5, 8, 9, and Fig. 3a. CLSM images revealed that DsD-1 of the *Desulfovibrionales* is the biomineralizing endosymbiont (Fig. 6a and Supplementary Videos 8, 9). This is further supported by genomics, as the DsD-1 genome is the only one containing magnetosome genes, assembled into a partial cluster composed of *mad* genes and some of the *mam* genes involved in magnetite formation in the *Desulfobacterota*[52] (Supplementary Fig. 7e and Supplementary Data 5). Mam proteins of DsD-1 share the highest sequence identity with their corresponding homologs in the reference strains *Solidesulfovibrio magneticus* RS-1[53], and *Fundidesulfovibrio magnetotacticus* FSS-1[54], with average identities of 54.4% and 51.8%, respectively. The 5 to 6 biomineralizing cells cluster with the *Desulfarculaceae* cells, which occupy most of the host's cytoplasm and extend over two-thirds of its length at the anterior pole. In contrast, the gammaproteobacterium GD-1 and alphaproteobacterium MD-1 lack specific spatial localization.

**All endosymbionts possess reduced genomes and have lost the ability to form flagella**
Genome sizes of the endosymbionts are 1.3, 2.0, 2.3, and 2.8 Mbp for MD-1, GD-1, DsD-1, and DcD-1, respectively (Supplementary Data 4). Despite its reduced size, the genome of DcD-1 has a relatively high G + C content (60.5%) compared to the other three genomes, whose G + C contents are significantly lower (from 37.2% to 45.7%, Supplementary Data 4). The average protein-coding density of GD-1 genome is within the typical range of 85–90% for free-living bacteria[55], whereas this density is lower for other endosymbionts, down to 57.6% for DsD-1, which contains the highest number of pseudogenes (>6%) compared to the three others (<1%).

Protein-coding genes were classified into Clusters of Orthologous Groups (COGs) using eggNOG-mapper v2.1.12[56] (Fig. 5e and Supplementary Data 6). The gene repertoires of the ciliate symbionts are typical of obligate endosymbionts. Six out of 22 COG categories detected once in all genomes were prominently represented in all

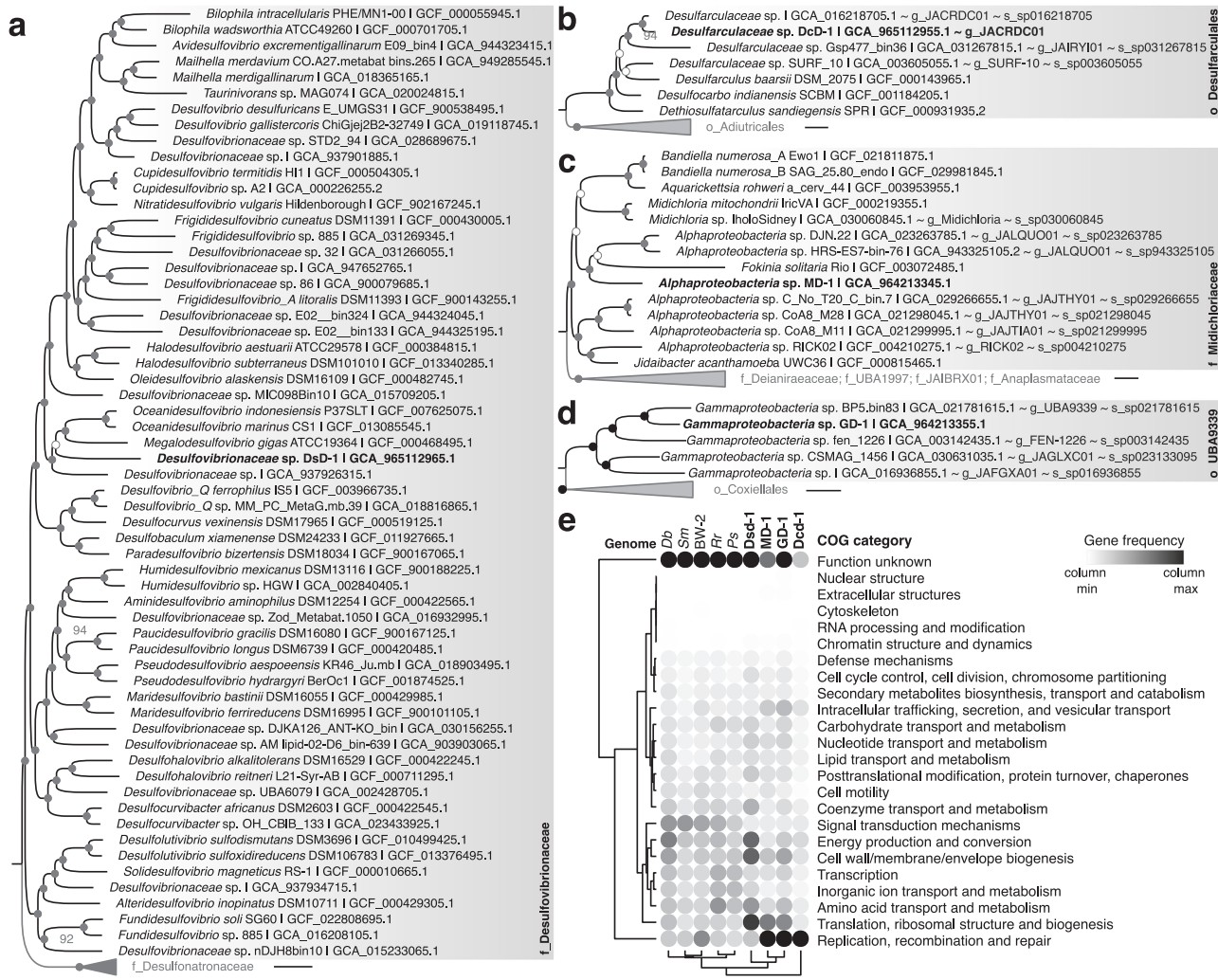

**Fig. 5 | Phylogenetic positioning of the four endosymbiotic bacteria present in the magnetotactic protist. a–d** Maximum-likelihood trees of the *Desulfovibrionaceae* family, *Desulfarculaceae* family, *Ca.* Midichloriaceae family and UBA9339 order (a subgroup of the *Legionellales* in *Gammaproteobacteria*) show relationships of DsD-1, DcD-1, MD-1, and GD-1, respectively, with the closest species for which a high-quality genome is available. Maximum-likelihood trees were built based on 120 conserved proteins used by the GTDB taxonomy. Branch lengths represent the number of substitutions per site. The black circles plotted on the internal nodes represent the statistical support, considered satisfactory, as the non-parametric bootstrap value was greater than 95% (estimated from 1000 replicas). White circles represent values between 70 and 95%. The endosymbiont genomes from this study are shown in bold. The corresponding Genbank accession numbers are given in the sequence names, along with the corresponding order name "o_", family name "f_", genus name "g_", and species names "g_" in GTDB. External groups for the *Desulfobacterota* and *Ca.* Midichloriaceae trees were identified based on the whole genome tree of Murali et al.[128] and Schön et al.[129], respectively. That of the UB9339 order was chosen based on the *Gammaproteobacteria* tree built in this study (the file in Newick format is given in Supplementary Data 3). Each external outgroup is composed of several genomes representing each genus/family. Phylogenetic trees based on 16S rRNA gene sequences are also given in the Supplementary Fig. 7. **e** Heatmap representing the functional classification of protein-coding genes classified by eggNOG-mapper version 2.1.12 into Cluster of Orthologous Groups (COG) for symbiotic species compared to a few free-living MTB and non-MTB species of the same phylum. The color shade represents the frequency of protein-coding genes for symbionts DsD-1, MD-1, GD-1, and DcD-1, and a few free-living species, including *Pseudomonas syringae* B728a (*Ps*), the magnetotactic gammaproteobacterium sp. BW-2 (BW-2), *Desulfarculus baarsii* DSM 2075 (*Db*), *Solidesulfovibrio magneticus* RS-1 (*Sm*), and *Rhodospirillum rubrum* ATCC 11170 (*Rr*). A relative color scheme uses the minimum and maximum values in each column to convert values to colors. Full classification and values are given in Supplementary Data 6.

endosymbionts aside "unknown functions"—ranging from 8 to 16% (Fig. 5e). Three of them are essential for cellular metabolism of primary endosymbionts[57], representing on average 16% of the genome: "Translation, ribosomal structure and biogenesis", "Amino acid transport and metabolism", and "Energy production and conversion". Two others—"Intracellular trafficking, secretion, and vesicular transport" and "Cell wall/membrane/envelope biogenesis"—accounted for 4-6% of the proteins. The COG category "Replication, recombination, and repair" represented 14%, 16%, and 30% of the predicted proteins in the endosymbionts GD-1, MD-1, and DcD-1, respectively. Of these, 46%, 63%, and 94% corresponded to mobile genetic element (MGE)–related genes. In contrast, such genes were rare in DsD-1, where this COG

category accounted for less than 0.02% of all annotated proteins. All symbionts have a minimal machinery involved in DNA replication, homologous recombination, base excision repair or mismatch repair. The set of conserved genes includes polymerase genes together with *dna*, *rec*, *ruv*, *uvr*, and *mut* genes (Fig. 5e and Supplementary Data 7). Functional annotation predicted genes encoding proteins with leucine-rich repeats (LRRs) and ankyrin repeats—domains involved in interactions with eukaryotic hosts[58]— in three out of four symbiont genomes (Supplementary Data 8). While the DsD-1, DcD-1, GD-1, and MD-1 genomes encode 0, 3, 8, and 7 ankyrin repeat-containing proteins, respectively, only GD-1 harbors four proteins with leucine-rich repeats, alongside several Type IV secretion systems (Supplementary

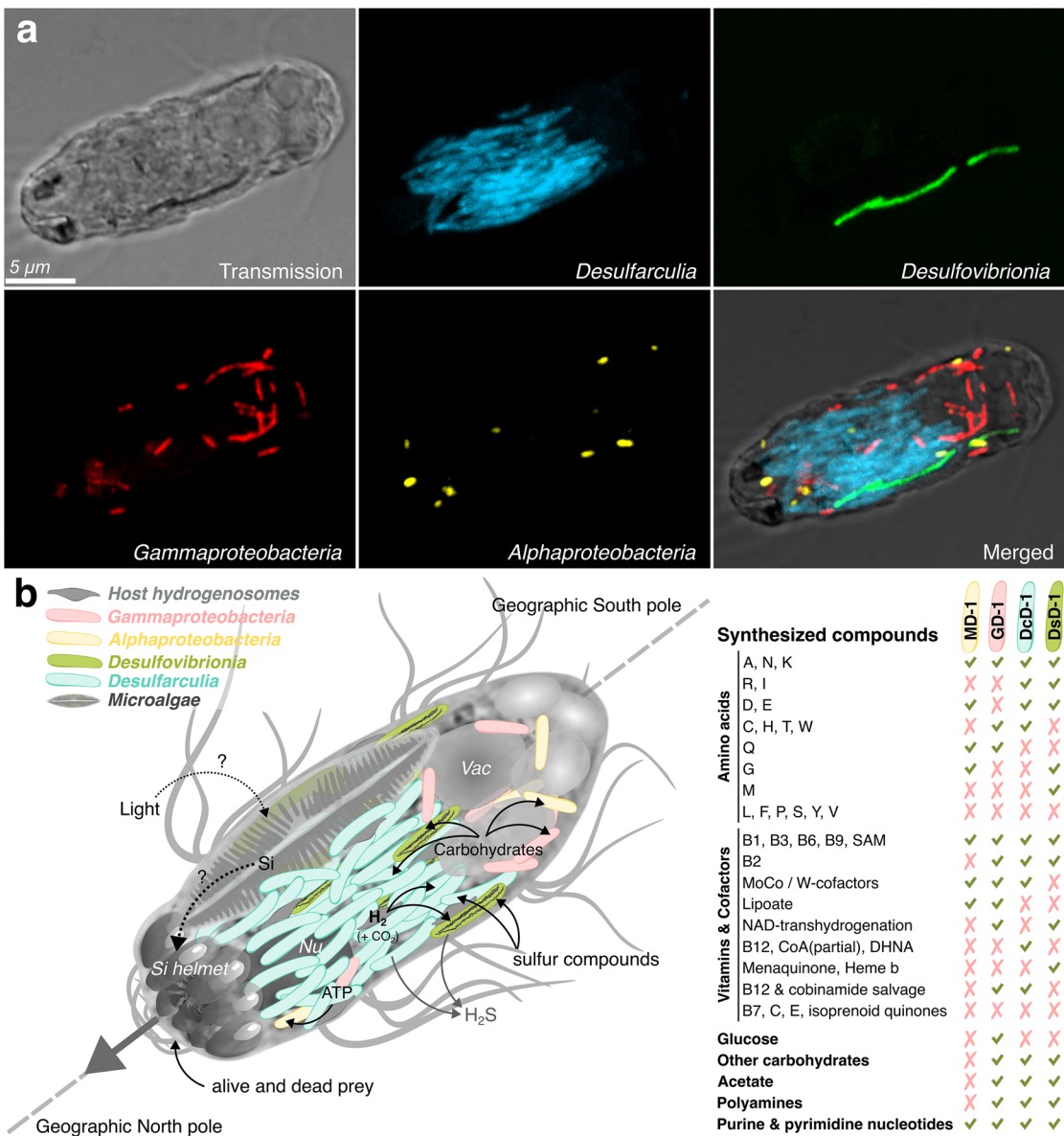

**Fig. 6 | Visualization and cellular organization of the four endosymbionts.**
**a** Fluorescent in situ hybridization (FISH) of the four endosymbiotic bacteria of a magnetotactic protist, validating their taxonomic affiliation and their localization in the host cell. The specific oligonucleotide probes were used to target the 16S rRNA gene sequence of the *Desulfarculia* (blue), the *Desulfovibrionia* (green), the *Gammaproteobacteria* (red), and the *Alphaproteobacteria* (yellow), corresponding to the DcD-1, DsD-1, GD-1, and MD-1 genomes. These CLSM images were obtained at the same focus corresponding to a z-stack located in the middle of the magneto-tactic protist. The 3D-reconstruction of the different z-stack (16 slices every 6 μm) of this magnetotactic protist is shown in Supplementary Video 8. **b** Schematic illustration of the magnetotactic consortium showing the magnetotactic behavior in the Northern Hemisphere and the syntrophic interactions between partners. The dotted line and the arrow show the anterior–posterior orientation of the organism relative to the Earth's magnetic field. The H₂, acetate, and CO₂ are produced following ATP synthesis by the hydrogenosomes present in the vicinity of *Desulfovibrionia* and *Desulfarculia* cells, and could be used by sulfate-reducers. Complex carbohydrates such as glucose would benefit the endosymbiotic *Gammaproteobacteria* and *Alphaproteobacteria* that could feed from the primary breakdown of organic material in digestive vacuoles. Symbionts are auxotrophs for some vitamins, cofactors, amino acids, and other compounds essential for growth. A summary of the symbionts' predicted biosynthetic capabilities based on their genomes is provided, with check marks (✓) indicating the presence and cross marks (✗) indicating the absence of each pathway. The vitamins are represented by one-letter code as follows: alanine (A), arginine (R), asparagine (N), aspartic acid (D), cysteine (C), glutamine (Q), glutamic acid (E), glycine (G), histidine (H), isoleucine (I), leucine (L), lysine (K), methionine (M), phenylalanine (F), proline (P), serine (S), threonine (T), tryptophan (W), tyrosine (Y), valine (V). MoCo stands for molybdenum cofactor, W-factors for tungsten cofactors, CoA for coenzyme A, and DHNA for 1,4-dihydroxy-2-naphthoate. Full metabolic network analysis is given Supplementary Fig. 8 and Supplementary Data 12.

Data 9), which are often repurposed by endosymbionts to deliver host-targeted effectors[58].

Conversely, COGs related to "Cell motility" and "Signal transduction mechanisms" are highly reduced in all symbionts compared to free-living counterparts (Fig. 5e). MacSyFinder v 2.1.2[59] did not predict the ability to form a flagellum in any of the four endosymbionts. Few flagella genes are present in MD-1 and DsD-1 (Supplementary Data 10).

No additional macromolecular system is encoded, except in the GD-1 genome, in which several secretion systems were predicted, in addition to the three type IV secretion system proteins (T4SS_TypeF_6, pT4SS_I, pT4SS_T)[60]. Other secretion systems include three type V secretion systems—including an autotransporter T5aSS and a two-partner system (T5bSS)—and a mannose-sensitive hemagglutinin (MSHA) pilus, which is a type IV pilus[61] (Supplementary Data 9). We

found 21 Dot/Icm genes (colocalized with the pT4SSi) involved in the virulence of *Legionella pneumophila* and *Coxiella burnetii*[62] (Supplementary Data 11). These genes are colocalized in the genome with the pT4SS$_i$, a secretion system that was suggested to be implied in host association in these lineages[63]. A total of 118 genes in GD-1 showing remote homology to known virulence factors were also predicted, including a *phtA*-like gene, a phagosomal transporter involved in the establishment and persistence of *L. pneumophila* within its replication vacuole[64], as well as enhanced entry (Enh) proteins that facilitate host cell invasion and infection[65]. Many transporters involved in iron uptake, adherence, and toxins (e.g., Enterobactin, Hemolysin, LPS, Microcin H47) in pathogens such as *Neisseria meningitidis* MC58, *Escherichia coli* O157:H7, *Haemophilus influenzae* Rd KW20, were also predicted. However, some sequence identities being low (i.e., 30-40%), these homologs may be functionally different. No such genes were detected in MD-1, except antophagocytosis genes, like the two other symbionts exhibiting similar but less numerous defensive systems (Supplementary Data 11).

### Metabolic network reconstruction of endosymbionts reveals hydrogen and sulfur-based syntrophies

Metabolic pathways were inferred using the MetaCyc database[66] and Pathways Tools software[67]. Besides revealing auxotrophy for several amino acids, cofactors, and vitamins, a common feature in many symbionts, the analysis indicates possible syntrophies based on the degradation of organic compounds such as acetate or glucose, the oxidation of hydrogen and sulfate reduction (Supplementary Fig. 8 and Supplementary Data 12). Together with previous analyses, these predictions helped us to develop a model describing the holobiont functioning (Fig. 6b).

Both *Desulfobacterota* symbionts, DcD-1 and DsD-1, have the potential to perform glycolysis and produce acetate and ATP from acetyl-CoA. They may also be able to reduce sulfate ($SO_4^{2-}$) to hydrogen sulfide ($H_2S$) using $H_2$ or organic substrates as electron donors. The *Desulfarculales* endosymbiont DcD-1, which accounts for most of the endosymbionts, may perform autotrophy. Indeed, we identified genes involved in the carbonyl branch (*acs* genes; EC 1.2.7.4), and the methyl branch (*folD*, *metF*, and *acsE* genes) (Supplementary Data 13) of the Wood−Ljungdahl pathway, which enables acetyl-CoA synthesis from $CO_2$ reduction in sulfate-reducing prokaryotes[68]. This partial prediction may result from incomplete genome assembly and requires further validation. No other canonical $CO_2$ fixation pathways were found in the four endosymbionts. The biomineralizing symbiont seems to play a central role in respiration and electron flux within the whole consortium, as only the Dsd-1 genome harbors genes encoding the synthesis of a low-potential menaquinone that is central to anaerobic respiration[69].

The GD-1 and MD-1 genomes show very different metabolic profiles. None of the two genomes encodes for cytochrome *c*-type *of cbb₃*-type oxidases or other enzymes associated with inorganic terminal electron acceptors (Supplementary Data 12). MD-1 has a severely reduced energy metabolism and lacks the ability to perform glycolysis as well as the tricarboxylic acid (TCA) cycle−similar to many *Rickettsiales*, which import metabolites from their host[70]. However, MD-1 has the potential to generate reducing power (NADH), acetyl-CoA, and $CO_2$ through the decarboxylation of exogenous pyruvate. It may also produce the (2S)-ethylmalonyl-CoA intermediate required for the ethylmalonyl-CoA pathway, an alternative to the TCA cycle. Notably, this endosymbiont is the only one in HBD-1 to encode three ADP/ATP translocases (*tlc;* InterPro entry IPR004667), which enable the direct import of ATP from the host cytosol[58]. In contrast, the genome of the *Gammaproteobacteria* GD-1 appears less reduced and is likely able to perform glycolysis through a variant pathway based on sucrose and glucose. Additionally, GD-1 could utilize the non-oxidative branch of the pentose phosphate pathway, which is absent in MD-1. Symbionts

harbour a repertoire of functions that sustain these relationships, including detoxification pathways for superoxide radicals, 8-oxo-(d) GTP (EC 3.6.1.55) or acidic resistance (EC 4.1.1.19) (Supplementary Data 13). MD-1 genome also encodes *ahpD* and *ahpC* genes (EC 1.11.1.28/1.11.1.26) involved in antioxidant defense in bacteria, specifically in protecting cells from oxidative stress caused by reactive oxygen species (ROS)[71].

## Discussion

A recent discovery of marine magnetotactic holobionts revealed a unique ability of certain microeukaryotes to navigate guided by the geomagnetic field[30] in anoxic marine sediments. This finding introduced the concept of collective magnetotaxis characterized by a mutualistic and obligatory symbiosis relying on interspecies hydrogen-transfer-driven syntrophy. The labor in magnetotactic holobionts is divided between partners: the euglenozoan host performs chemo-aerotaxis, while the ectosymbiotic bacteria provide magnetoreception. Although freshwater MHBs are composed of a totally different consortium and organizational pattern, they rely on the same collective magnetotaxis. While marine MHB involve a two-way partnership between an Excavata host and epibiotic *Desulfobacterales*, freshwater MHB represent a more complex consortium. This consortium was retrieved from anoxic sediments at several locations, including the Dordogne River, the Cère River, Lake Aydat, as well as a pond and a stream in Britany. This consortium includes a distinct eukaryotic host and diverse bacterial endosymbionts, one of which biomineralizes magnetosomes. None of the organisms in this freshwater consortium corresponds to previously described species or genera.

The host is a ciliate (*Ciliophorea*) with synapomorphies typically encountered in ciliates of the *Prostomatea*, but exhibits few apomorphies (i.e., derived or novel characters) compared to known *Prostomatea* and *Plagiopylea* species, from which it is genetically divergent. These differences suggest the need for classification into a new taxonomic group. The 18S rRNA gene sequence might not be resolutive enough to get a proper insight into the relationships with other ciliates. The tree topology lacks support and differs from those reconstructed from numerous orthologs[72]. A more resolved phylogenetic reconstruction was not possible as the genome of the host could not be assembled from the amplified DNA of the holobiont. The most distinctive host feature is a "silica helmet" surrounding the anterior oral opening, a structure consistently observed in all freshwater MHB across various environments. The magnetotactic protist HBD-1 harbors four symbiotic bacterial species represented by GD-1, affiliated with the *Gammaproteobacteria*−belonging to an unknown order close to the *Legionellales* and *Coxiellales*−MD-1, affiliated with the *Rickettsiales* order from the *Alphaproteobacteria* class, and DcD-1 and DsD-1 affiliated with the *Desulfobacterota* phylum. These two sulfate-reducing bacteria belong to divergent groups: the DcD-1 genome affiliates with the *Desulfarculaceae* family, and that of DsD-1 with the *Desulfovibrionaceae* family, the latter biomineralizing bundles of bullet-shaped magnetosomes. While endosymbiotic lifestyles (e.g., parasites or mutualistic symbionts) are well-documented in the *Pseudomonadota* groups[73,74], there are currently no widely documented cases of *Desulfarculaceae* or *Desulfovibrionaceae* as true intracellular symbionts of protists[75,76]. Most species in these groups are typically found in anoxic environments such as aquatic sediments and animal guts[77]. When they are involved in symbioses, they are external epibionts[39]. Given the novelty, we propose that genomes of GD-1, MD-1, and DsD-1 serve as the nomenclature type for the species *Protisticella dordonia*[TS], *Midichloriella endociliophora*[TS], and *Endodesulfobacter magneticum*[TS] according to the rules of the SeqCode[78] (Supplementary Results 1). As the genome quality of DcD-1 does not meet the SeqCode criteria, we assigned *Candidatus* status to the proposed name *Desulfella intracellularis* for the species represented by this genome.

The ultrastructure of *Ca*. Desulfarcum epimagneticum CR-1[30] and *Endodesulfobacter magneticus* DsD-1, as well as their physical interactions with their hosts, are different in marine and freshwater MHB. It raises questions about how these differences may impact the functioning of collective magnetotaxis. In marine MHB, approximately 150 *Ca*. D. epimagneticum cells form an epibiotic layer on the euglenozoan host. Each cell produces a chain of 27 rhomboidal dodecahedron-like magnetite particles, resulting in a total of about 4000 magnetosomes per holobiont[30,32]. The number of magnetosome chains far exceeds the minimum number of a few dozen magnetosomes required to generate sufficient torque in such MHB to align with Earth's magnetic field lines[32]. Conversely, HBD-1 is larger than the marine MHB but harbors 25 times fewer endosymbiotic cells. In total, approximately 800 magnetosomes are produced by *E. magneticus* DsD-1 cells from a single host to induce a magnetotactic behavior in the ciliate. Despite their lower numbers, each cell produces almost five times more magnetosomes aligned into six chains of bullet-shaped magnetite crystals. These particles are nearly twice as large as those of free-living magnetotactic bacteria from the *Desulfovibrionaceae* family, such as *Solidesulfovibrio magneticus* strain RS-1[79] or *Fundidesulfovibrio magnetotacticus* strain FSS-1[54], which typically form about ten particles of similar width but less than 100 nm in length. However, they are smaller than those in a magnetotactic eukaryote described by Leão et al. [35], which had magnetosome-like particles averaging $276.6 \pm 61.3$ nm in length and $52.7 \pm 5.1$ nm in width. Another striking difference between both magnetosome-producing symbionts is that the endosymbiont apparently evolved to overproduce magnetosomes to the point that they fill the entire cytoplasm. Indeed, cells of DsD-1 produce about 131 particles for a length and width of approximately 5 μm and 0.5 μm, whereas cells of the cultivated strain FSS-1 produce about 10 magnetosomes for a length and width of approximately 4 and 0.8 μm, respectively[54].

Such endosymbionts were not observed in the magnetotactic organism described by Leão et al. [35], who emphasized that magnetite crystals were biomineralized by the protist without any further assistance of endosymbionts. It can be questioned whether this microeukaryote belonged to the same MHB described here. Indeed, it has the same morphology, size, cilia, and the contrasted structures at the anterior pole associated with the quartz inclusions. Re-examination of Leão et al.'s TEM images (inset of Fig. 1D[35]) reveals contrasted structures surrounding the magnetosome chain bundles. These structures may correspond to endosymbiotic magnetosome-forming cells, as they show the same contrast as the one observed around magnetosomes in the present study. Although magnetosomes appear larger (i.e., 92% for the length and 29% for the width) than those in our study, additional evidence supports this hypothesis: the number and organization of magnetosome bundles, as well as the magnetosomes' shape and number, are similar. Unfortunately, limited biomass (i.e., one cell per liter of porewater) and irregular organism occurrence hindered in-depth characterization of the magnetotactic protist in Leão et al.'s study[35]. We attempted further ultrastructural and genomic analyses of magnetotactic protists from the Ururaí River and Ubatiba River in Rio de Janeiro, Brazil. Due to challenges with sampling, we were only able to obtain microscopy images (Supplementary Fig. 9) and videos (Supplementary Video 10). In the STEM-HAADF images (Supplementary Fig. 9b), the contrasted area surrounding the magnetosome chains is more pronounced compared to conventional TEM images (Supplementary Fig. 9a). Furthermore, video microscopy of fresh samples, which contained a high number of microorganisms exhibiting magnetotaxis, revealed the magnetosome chains located near the oral bulge at the protist's apical position. Our observations strongly suggest that both protists from France and Brazil are magnetotactic thanks to biomineralizing endosymbionts. Magnetite biomineralization by microeukaryotes still needs to be confirmed.

The association of protist hosts from diverse phyla with symbionts from various classes suggests that magnetosymbiosis has independently emerged multiple times across both the *Bacteria* and *Eukaryota* domains. Moreover, the emergence of these symbioses will likely be facilitated in environments where magnetotactic prokaryotes are prevalent and common. This suggests that environmental pressures in oxygen-deprived aquatic habitats might commonly favor the development of such symbiotic relationships among microeukaryotes. However, the strict anaerobic lifestyle of the ciliate host remains in question. Indeed, although it lives below the oxic-anoxic interface, it is sensitive to air, contains putative hydrogenosomes, and has established symbiosis with anaerobic bacteria; we are missing genomic information that would confirm this point. Moreover, the presence of microalgae raises questions about how these can all sensibly work together. The specific type of magnetosomes seems less critical than the overall magnetic moment they provide. Despite structural and taxonomic differences between MHB, certain metabolic functions appear redundant across both symbionts and hosts. These functions seem to have been selected independently in both cases, highlighting convergent evolution likely driven by similar ecological or physiological demands. Many anaerobic protists, such as certain ciliates and flagellates, rely on hydrogen-consuming prokaryotic endosymbionts, often methanogenic archaea or sulfate-reducing bacteria[80]. Here, both magnetotactic protists appear to rely on these syntrophies. Predatory protists metabolize organic matter and produce hydrogen ($H_2$) as a byproduct via their hydrogenosomes and small molecules such as $CO_2$, acetate, or alcohols. In the magnetotactic ciliate, these intermediates are likely further oxidized by their endosymbionts for biomass and ATP generation. On the other hand, the host could access energy produced by its sulfate-reducing endosymbionts DcD-1 and DsD-1. Such utilization of symbionts as electron acceptors or sinks in metabolic coupling has been previously documented in other ciliates[81]. In addition, hydrogen sulfide produced by sulfate respiration of DcD-1 and DsD-1 could help maintain a low redox potential inside the consortium or could be eliminated outside the protist by diffusion to avoid toxic effects. As for magnetotaxis, the division of labor enhances metabolic efficiency for both host and symbionts in anoxic sediments, linking organic matter decomposition to broader biogeochemical processes.

While the benefits and interactions with both *Desulfobacterota* DcD-1 and DsD-1 can be predicted, those involving the two *Pseudomonadota* MD-1 and GD-1 are less clear. They represent two previously undescribed species and genera, each forming a monophyletic group with other intracellular symbionts of metazoans and a few protist lineages. The GD-1 genome affiliates with the *Legionellales*, a group composed of host-adapted, intracellular bacteria, including the accidental human pathogens *Legionella pneumophila* and *Coxiella burnetii*[82]. A growing number of environmental surveys indicates that the members of this group are diverse and are endosymbionts of various hosts, including arthropods and microeukaryotes such as amoebae[49,63]. Recent ecological surveys have uncovered an unexpected diversity of *Legionellales* associated with non-vertebrate hosts and harbor a complex mutualistic lifestyle, although such diversity in microeukaryotes remains largely undocumented. The genome of *Protisticella dordonia* GD-1 contains numerous homologs of virulence factors described in the invasion of unicellular eukaryotes[62]. However, depending on their impact on host fitness, the same virulence factors can be involved in a mutualistic lifestyle too. They enable resistance against grazing as well as colonization of digestive vacuoles or phagocytosomes, which is an unconditional step towards the establishment of mutualistic endosymbiosis[63]. The alphaproteobacterium MD-1 belongs to an equally undiversified and little-studied family within the *Rickettsiales*: the *Midichloriaceae* family, which is better known for its mutualistic way of life within unicellular eukaryotes[46–48]. Genomics suggests that endosymbiotic *Pseudomonadota* MD-1 and GD-1 rely on organic substrates produced by their host, and that MD-1 additionally benefits directly from host-derived ATP. They likely do not participate

directly in the global energy metabolism of the MHB and seem to be unable to synthesize a large number of compounds required for their growth. However, they may support the consortium by synthesizing essential compounds for both the host and other symbiotic partners, such as a few vitamins and key cofactors. Similar observations were done in the *Gammaproteobacteria* Comchoano, symbiont of the marine choanoflagellate *Bicosta minor*[63].

The association of GD-1 and MD-1 with their ciliate host appears ancient, as their genome structures reflect an advanced stage of reductive evolution. However, further investigation is required to date the relative emergence of this symbiosis in their respective taxonomic groups. Indeed, although families of the *Rickettsiales* order are considered to be relatively old groups of intracellular bacteria[58,83,84], associations with particular hosts may have arisen more recently. On the other hand, UBA9339 symbiont resides in a poorly explored group of orders related to *Legionellales* and *Coxiellales*. While there are suggestions that host association in general may be ancestral in these lineages, even its obligatory nature may be a more recent trait[63]. Despite its relatively large genome size (>2 Mbp), the magnetic symbiont DsD-1 shows clear signatures of a long-term obligate symbiotic lifestyle as well. DsD-1 contains very few mobile genetic element–related genes and almost no pseudogenes, and it has a minimal set of conserved genes for DNA replication, recombination, and repair[55]. However, in such advanced reductive evolutionary processes, one would not expect the unusually low protein-coding density observed in DsD-1 (57.43%), which indicates that nonfunctional sequences have not yet been fully purged. This genomic architecture suggests that DsD-1 may be undergoing an advanced, deletion-driven stage of genome reduction that followed the earlier loss of MGEs and pseudogenes. In contrast, the symbiosis establishment could have been more recent for the *Desulfarculales* endosymbiont DcD-1, whose genome is highly pseudogenized (6 %) and contains more than 1.5 k MGEs. Such a genomic structure could be the result of, or a remnant of, ongoing reductive evolutionary processes following endosymbiosis establishment, as observed in other symbiotic bacteria[55,75,85–87].

Genomic analysis provides compelling evidence that the bacterial endosymbionts of HBD-1 exhibit an unequivocal intracellular lifestyle. However, the same level of certainty does not apply to the intracellular diatoms and other microalgae almost always observed in freshwater environments where MHB are found. The presence of different diatom species within the same population of magnetotactic ciliates from a single sample supports the idea that intracellular diatoms are not essential for the functioning of the ciliate or the holobiont. Nevertheless, additional evidence is needed to clearly define the nature of this relationship. The exact process by which diatoms and other microalgae are internalized remains unclear, as direct observations are lacking. Because diatoms and protists are similar in size, and given the small diameter of the cytostome opening (0.5 μm), it is unlikely that the cytostome can expand enough to engulf the entire rigid silica frustule of diatoms, as also observed in other ciliates such as *Didinium nasutum*[88]. Moreover, the silica composition of the diatom frustule may further limit its flexibility. Alternatively, diatoms and other algae may be phagocytosed through the plasma membrane of the ciliate and subsequently internalized within phagosomes[89]. As in *D. nasutum*, the pores of HBD-1 could be involved in the fluid removal after prey engulfment[88].

The presence of intracellular silica-rich granules is intriguing, especially in terms of their localization, origin, and their role in ciliate biology. To date, similar Si-rich granules have been observed in one ciliate species only, *Maryna umbrellata*[87], which belongs to the *Colpodea* (i.e., a divergent clade to *Prostomatea*). Foissner et al.[90] showed evidence that the ciliate was able to biomineralize amorphous silicon (glass) $[(SiO_2)]_n$ granules. In *M. umbrellata*, it was hypothesized that glass beads could be involved in light perception and protection against mechanical stress and predators[90]. Interestingly, when *M.*

*umbrellata* enters the dormant stage, most glass granules are excreted to form the surface cover of the globular resting cyst. However, the intracellular organization, size, and shape of these granules in the magnetotactic ciliate differ significantly from those in *M. umbrellata*. Indeed, the granules of HBD-1 are crystalline, specifically localized in the frontal part of the protist, of various sizes (1–5 μm) and shapes, whereas *M. umbrellata* produces amorphous ellipsoidal granules with an average size of $819 \times 630$ nm localized in the cytoplasm of the trophic cells. Here, additional analyses are required to confirm whether quartz granules are biomineralized by the ciliate. Another scenario would be that these minerals are environmental detrital quartz grains engulfed by phagocytosis. The density of those inclusions being higher than the density of the cytoplasm, they may induce a gravitropism. Thus, in addition to the roles proposed previously (i.e., Light perception and protection against mechanical stress and predators), we propose that these granules could provide additional assistance in the vertical navigation within the chemical gradients of their environment, complementing the guidance offered by magnetotaxis.

In conclusion, this study reports the discovery of a eukaryotic organism capable of sensing the geomagnetic field through endosymbiotic integration, expanding our knowledge of the diversity of magnetosymbioses in aquatic habitats. It also broadens our understanding of the diversity of biomineralization processes. The magnetotaxis observed in the magnetotactic holobiont HBD-1 from the Dordogne river could mirror that of free-living magnetotactic bacteria and euglenozoan, enabling efficient navigation in chemically stratified aquatic environments. Considering the current evolutionary scenarios of eukaryogenesis and eukaryotes diversification[9], it is appealing to link these findings to the origin of magnetoreception in eukaryotes more largely. However, although the integration of magnetotactic bacteria into a modern lineage of protists is now proven, further research is needed to determine whether such integration and further organellogenesis or lateral gene transfers could have occurred in early holobionts that gave rise to the first eukaryotes lineages able of magnetoreception.

## Methods

### Site description and sample collection
Sediment samples were collected from the shore of the river Dordogne at Beaulieu-sur-Dordogne, Nouvelle-Aquitaine, France (44.9783°N, 1.8378°E) at different seasons between August 2021 and January 2023. Samples were collected in 1 L bottles filled with one-third of sandy sediments and completed to full capacity with water from the site to exclude air bubbles. Samples stored at 10 °C with the cap slightly opened allowed the conservation of magnetotactic protists for more than 3 months.

### Magnetic enrichment and light microscope observation
Prior to the concentration, chemical gradients of the microcosm were broken, and the sediments were resuspended by mixing the bottle. Magnetotactic organisms were concentrated by placing the south pole of a magnetic stirring bar next to the sample bottle above the water-sediment interface for 1 h. Magnetically concentrated cells were collected and observed using the hanging drop assay[91] with a ZEISS Primo Star light microscope equipped with a phase-contrast optic.

### Mesocosms measurements of chemical and cell count profiles
Dissolved oxygen concentration profiles were measured in the water column and sediments of the microcosms using a fiber-optic oxygen sensor (50-μm tip diameter, REF OXR50) and a FireStingO2 meter, both from Pyroscience (Germany). High-resolution profiles (100 μm steps, from 10 mm above the sediment to −30 mm below the sediment) were achieved with a Pyroscience MU1 motorized micromanipulator. Sensor calibration was made with a 50 mL milliQ water solution flushed with air for 30 min ($O_2$ saturation = 100%) and a milliQ water solution

flushed with N₂ for 30 min (O₂ saturation = 0%). Magnetotactic protists were not sufficiently concentrated to carry a cell count profile. Therefore, sediment was harvested every 1 cm after oxygen profile measurements and transferred into a 50 mL Falcon™ tube containing filtered water from the Dordogne River. A magnetic concentration was then performed for 1 h, and magnetotactic organisms were observed using the hanging drop assay. Magnetotactic protists were always observed in regions depleted of oxygen, i.e., below 1 cm of sediment. Measurements were systematically carried out on three replicates of three different samples.

### Transmission electron microscopy (TEM)

TEM was used to analyze the ultrastructure of the magnetic protists that were directly deposited onto TEM copper grids coated with a carbon film and with poly-L-lysine. Electron micrographs were acquired with a Tecnai G$^2$ BioTWIN (FEI Company) equipped with a CCD camera (Megaview III, Olympus Soft Imaging Solutions GmbH) operating at an accelerating voltage of 100 kV. High-resolution TEM (HRTEM) was performed using a JEOL 2100 F microscope. This machine, run at 200 kV, was equipped with a Schottky emission gun, an ultra-high-resolution pole piece, and an ultrathin window JEOL XEDS detector. HR-TEM images were recorded using a Gatan US 4000 CCD camera.

Thin-sectioned samples were obtained from magnetically concentrated protists fixed in 0.25% (w/v) glutaraldehyde in sodium cacodylate buffer (10 mM, pH 7.4) and stored at 4 °C for at least 24 h. Cells were subsequently washed with the same buffer solution and post-fixed for 1 h with 1% (w/v) of osmium tetroxide in cacodylate buffer. Magnetic cells were then dehydrated through a graded series of ethanol solutions with increasing concentrations and finally embedded in Epon 812 monomeric resin. All the chemicals used for histological preparation were purchased from Electron Microscopy Sciences. Sections (80 nm thick and 3 mm long) were cut using an ultramicrotome UC7RT (Leica Microsystems GmbH), mounted onto TEM copper grids, and stained with uranyless and 3% lead citrate (Reynolds Lead Citrate, Uranyless) for 3 min.

Magnetosome and endosymbiotic cell sizes were determined from TEM images using ImageJ software (v 1.48). Several cells were used to calculate the mean value and standard deviation for each feature.

### Scanning electron microscopy (SEM)

SEM was employed to examine the preserved morphology of the magnetotactic protist. Magnetically concentrated magnetic protists were (i) micromanipulated and chemically fixed with 0.25% glutaraldehyde on a glass coverslip coated with poly-L-lysine, (ii) dehydrated through a graded ethanol series (50%; 70%; 96%; 100%), (iii) dried through critical point drying (CPD) (Leica EM CPD300), and finally (iv) coated with carbon using a Leica EM SCD500 carbon coater. Observations were carried out using a Zeiss Ultra 55 field emission gun SEM. Secondary electron images were collected with an InLens detector and an accelerating voltage of 1–3 kV with a working distance of 3.5 mm. Backscattered electron imaging (AsB detector) and elemental mapping by EDXS were performed at accelerating voltages between 10 and 15 kV with a working distance of 7.5 mm.

### Synchrotron-based nanobeam-scanning X-ray fluorescence (nano-XRF)

Sorted and 0.25% glutaraldehyde fixed magnetotactic protists were deposited on silicon nitride Si₃N₄ membrane for measurement at the ID16B hard X-ray nanoprobe beamline (ESRF, Grenoble)[92]. Nanobeam-scanning X-ray fluorescence (nano-XRF) data were collected under ambient conditions using an incident photon energy of 17.5 keV with 11 Si-drift detectors to measure X-ray fluorescence (XRF) ~15 degrees to the sample in the forward scattering direction. The focused X-ray beam was measured to be 55 × 56 nm (FWHM) before XRF mapping. A raster scanning step size of 50 nm was used with a dwell time of 50 ms to collect high-resolution XRF maps. XRF spectra were energy calibrated and fitted with Kα emission lines of second/third row periodic elements (e.g., Si, P, S, Ca, Fe, Cu, and Zn) using pyMCA v 5.9.2.

### Cryo soft X-ray tomography (cryo-SXT)

Imaging was conducted at ALBA synchrotron using cryo transmission X-ray microscopy at Mistral beamline (Barcelona, Spain)[93] under the awarded proposal number 2022025566. Using a similar approach to that described above, magnetotactic cells were magnetically concentrated on the wall of an environmental sample bottle and then extracted by micropipette. Micromanipulated magnetotactic protists were then fixed with 0.25% of glutaraldehyde for 10 s and deposited on a poly-L-lysine-coated transmission electron microscopy grid (Quantafoil R2/2 holey carbon, gold) along with 1 μL of 100 nm Au nanoparticles (BBI Solutions concentrated 5×). Gold nanoparticles deposited on the grid were used as fiducial references for projection alignment prior to tomographic reconstruction. The grid was incubated horizontally for 1–2 min to facilitate the settling of protists onto its surface. The grid was subsequently mounted vertically into a Leica EM GP plunge freezer at 95% humidity, blotted from the back of the grid with a filter paper for 3 s, and then rapidly plunged into a liquid ethane maintained at −180 °C, cooled by liquid nitrogen. Ice-vitrified cells were maintained under cryogenic conditions until transfer to the Mistral beamline cryo chamber for analysis.

A tilt series of projections from −65° to +65° was collected every 1° with an incident X-ray energy of 520 eV. Exposure time varied from 1–2 s for each projection (2 s at higher angles). The sample was imaged at 0° before and after collecting the tilt series to ensure there was no significant beam damage at the achievable resolution. A 40 nm Fresnel zone plate was used with an effective pixel size of 12 nm. The projections were normalized with the incoming flux and deconvolved with the measured point spread function (PSF) of the optical system[94]. Alignment of projections was done with Etomo using Au fiducials of 100 nm. Tomographic reconstruction and SIRT deconvolution were performed using IMOD. Although this approach conserves the organization of the magnetotactic protists in their near native-state configuration, they are slightly flattened due to their thickness. Nevertheless, the overall ultrastructure of the magnetotactic protists was maintained.

### Cell sorting and whole genome amplification (WGA)

Cell sorting was performed on sediment samples collected in the Dordogne River with an InjectMan® 4 micromanipulator and a CellTram® 4r, hydraulic, manual microinjector from Eppendorf mounted on a ZEISS Axio Vert.A1 KMAT microscope. The microscope and micromanipulator were installed inside a clean chamber, previously sterilized by 1 h germicidal ultraviolet irradiation (UV lamp 2 × 15 W, 254 nm). A 10-μL drop containing magnetically concentrated cells was gently added to a 30-μL drop of filtered Dordogne River water on a hydrophobic coverslip to enable magnetic migration of magnetotactic cells toward the filtered water. One magnetotactic protist was transferred with a sterile microcapillary (Drummond Scientific 3-000-203-g/X) into a 4-μL droplet of phosphate buffer saline (PBS). This droplet was stored at −20 °C prior to WGA. To generate sufficient gDNA for 16S rRNA gene and shotgun metagenomic sequencing, WGA was conducted using multiple displacement amplification (MDA) with the REPLI-g Single Cell Kit (QIAGEN), according to the manufacturer's protocol. The concentration of double-stranded gDNA was quantified using a QuBit™ 4 fluorimeter (ThermoFisher Scientific).

### Cloning and sequencing of the 18S rRNA/16S rRNA gene sequences of magnetically concentrated and sorted cells

The 18S and 16S rRNA genes were used to identify magnetically-concentrated eukaryotic and prokaryotic cells, respectively, from samples collected in the Dordogne River, an unnamed spring, and Lake Lannénec in Britany, Lake Aydat in Auvergne, and the Cere River a few

kilometers before it flows into the Dordogne River. DNA was amplified using the Phusion® Hot Start Flex DNA Polymerase following the manufacturer's instructions. Two sets of primers were used: for the 18S rRNA gene, EukA 5'-AACCTGGTTGATCCTGCCAGT-3' and EukB 5'-TGATCCTTCTGCAGGTTCACCTAC-3'[95], and 27 F 5'-AGAGTTT-GATCMTGGCTCAG-3' and 1492 R 5'-TACGGHTACCTTGTTACGACTT-3' for the 16S rRNA gene[96]. Blunt-end fragments of these gene sequences were inserted into vectors using the Zero Blunt® TOPO®PCR Cloning Kit with One Shot® TOP10 chemically competent E. coli cells. The resulting cloned inserts were submitted for sequencing (Eurofins Genomics Germany GmbH). Obtained sequences were aligned against the NCBI nucleotide database with Basic Local Alignment Search Tool[97]. The rRNA gene sequences were screened for chimeric structures using the UCHIME2 algorithm[98] and subsequently deposited in GenBank under accession numbers (see Data availability).

## Confocal laser scanning microscopy and fluorescence in situ hybridization (FISH)

FISH was performed according to Pernthaler et al. [99]. Specific probes for the four different endosymbiotic bacteria were designed based on the alignment of the most similar 16S rRNA gene sequences and labeled with different fluorochromes. Probe specificity was assessed using the PROBE_MATCH tool available in the RDP-II[100]. The nearest non-target sequences displayed at least one mismatch with the designed specific probes. The oligonucleotide probes used in this study were ordered from Eurofins Genomics (Supplementary Table 1).

Micromanipulated magnetotactic protists were deposited on a SuperFrost Plus® Gold, Menzel-Gläser (Thermo Scientific) microscope slide. Cells were fixed during 2 h with 30 μL of a solution of 1% paraformaldehyde. Fixed cells were then dehydrated by serial incubation for 5 min each in 50%, 80% and 100% ethanol. The hybridization solution contained 10 ng/ml of the probe, 0.9 M NaCl, 20 mM Tris-HCl (pH 7.4), 1 mM $Na_2EDTA$, and 0.01% sodium dodecyl sulfate (SDS), using the hybridization and washing stringencies of 30% for all probes (i.e., 0.6 mL of formamide per mL of hybridization buffer). Hybridization was performed at 46 °C for 2 h. Slides were mounted with a Pro-Long™ Diamond Antifade Mountant (Invitrogen™) with or without 4,6-diamidino-2-phenyl- indole (DAPI) and then were immediately observed with the Zeiss LSM980 confocal laser scanning microscope. The four specific probes HBD1-ALPHAp, HBD1-GAMMA1-2p, HBD1-DARCU1-2p, and HBD1-DARCU1-2p were designed with four different fluorochromes Alexa555, ATTO633, ATTO425, and ATTO488, respectively, to be able to distinguish the different cells and localize them. The specificity of each of the four probes was tested using the usual controls with DAPI and the EUBp. The specific probe HBD1-GAMMA1p with the fluorochrome ATTO488 was used for these tests to distinguish between the fluorescence of all bacteria at 633 nm (EUBp-ATTO633) and the endosymbiotic Gammaproteobacteria.

The multiplex labeling with ATTO425, ATTO488, Alexa555, and ATTO633 dyes were excited simultaneously with 4 lasers: 405 nm (0.3% power), 488 nm (0.7% power), 561 nm (1% power), and 639 nm (0.3% power). Images were acquired using the spectral mode (with MBS filters 405/488/561/639), and the signal was collected from 411 to 694 nm using a GaSP spectral detector. Reference emission spectra were recorded for each dye (using their corresponding laser excitation and MBS filters 405/488/561/633). In addition, a reference spectrum corresponding to autofluorescence in the cell was also recorded on unstained samples simultaneously with all laser excitation (405, 488, 561, and 639) and using the MBS filters 405/488/561/633. A complete set of these reference spectra was recorded. Spectral separation using these reference spectra was applied to multiplex images with an online-fingerprinting method, resulting in four separate dye channels and one autofluorescence channel. Images acquired in spectral mode represent the maximum intensity projection of a subset of 16 axial planes.

The image (Fig. 4c) acquired in Airyscan SR (Super-Resolution) mode using the Airyscan 2 detector was processed using the Airyscan joint deconvolution (jDCV) plugin of the ZEN blue software to enhance image resolution and clarity. The image was acquired in 16-bit mode and processed using Zen bleu software (Zen 3.6) and ImageJ (ImageJ2, version 2.16.0/1.54).

## Shotgun metagenomic sequencing, assembly, and functional annotation

The genomic DNA of the holobiont was sequenced at Genoscope with a combination of Illumina and Oxford Nanopore Technologies. First, 3 μg of amplified DNA was treated with T7 Endonuclease I (New England Biolabs) to resolve branches due to the multiple displacement mechanism. For Illumina sequencing, 250 ng DNA was sonicated to a 100–1000 bp size range using the E220 Covaris instrument (Covaris, Inc.). The fragments were end-repaired and 3' adenylated, and NEXT-flex HT barcodes were added (BioScientific Inc.). The ligated product was amplified using 12 PCR cycles with the Hifi HotStart ReadyMix kit (Kapa Biosystems) and purified with 0.8× AMPure XP reagent. After library profile analysis using the Agilent 2100 Bioanalyzer and quantitative PCR (qPCR) (MxPro; Agilent Technologies), the library was sequenced on an Illumina NovaSeq 6000 instrument. A total of $1.20 \times 10^8$ paired-end reads were obtained. The Illumina reads were trimmed by removing low-quality ($Q < 20$) nucleotides, sequencing adapters, and sequences of <30 nucleotides (nt), using the FastX-Toolkit package.

For Nanopore sequencing, 1 μg DNA was used for the library preparation following the 1D Native barcoding genomic DNA protocol with the EXP-NBD104 and SQK-LSK109 ligation kit (Oxford Nanopore). The library was sequenced using a Nanopore R9.4.1 revD flow cell and the PromethION device with MinKNOW v.4.5.4 and Guppy v.5.1.13 + b292f4d13 software. A total of 7,444,240 reads were obtained with an N50 value of 4.97 kb.

A hybrid assembly was launched using SPAdes v.3.13.0 with −sc option[101]. This assembly resulted in 280,856 contigs for a total length of 157 Mb. First, EukRep[102] was used to identify eukaryotic contigs within the assembly. Then all reads were mapped to these contigs with Bowtie 2[103] to estimate the number of eukaryotic reads. Contigs longer than 1000 bp were used as input for the Anvi'o metagenomic workflow[40] to perform the metagenomic binning based on tetra-nucleotide frequency. The binning resulted in four Bins corresponding to four bacterial SAGs (Supplementary Data 4). The automatic annotation of the different SAGs was performed with the MicroScope platform (https://mage.genoscope.cns.fr/microscope)[104]. Pseudogenes were detected by aligning each symbiont CDS to its ortholog in E. coli K-12 substr. MG1655 and identifying orthologs with disrupted structure. A gene was classified as fragmented when a single E. coli CDS matched two colinear ORFs in the symbiont genomes, indicative of frameshift mutations or premature stop codons. Assembly completeness, contamination, and redundancy of the different SAGs assemblies were estimated using CheckM2 v.1.0.2[105] with the "allmodel" option leading to the selection of the general model, and with anvi'o using anvi-estimate-genome-completeness tool with the defaults HMM collections of single-copy core genes named Bacteria_71[106]. The rRNA genes were predicted using RNAmmer[107] in all the global assemblies before binning, and we looked if the predicted 16S/18S rRNA belonged to a binned SAG. Then we performed a BLASTN[108] comparison to the rRNA / ITS NCBI database, selecting the 16S rRNA from Bacteria and Archaea type strains.

## Genome-based taxonomic classification and molecular phylogeny

We used the GTDB-Tk v2.1.1[42] software to assign a taxonomic classification to each bacterial SAGs. We could notice that the taxonomic classification was mostly defined by topology, suggesting that all our SAGs correspond to new bacterial genera and species.

The taxonomic position of each SAG was confirmed by building phylogenetic trees and the 120 (bac120) phylogenetically informative markers[42] used by GTDB. First, all genomes of good quality (i.e., >90% complete with <5% redundancy according to checkM values on GTDB) were downloaded from the Genbank database[109] in April 2024, and rapidly re-annotated with PROKKA v1.14.6[110] to homogenize coding sequence predictions. The number of high-quality genomes being scarce for some taxa, we lowered our stringency for genome completion to 80% for some of them. The conserved markers were extracted using PyHMMER 0.8.0[111] and the Hidden Markov Models (HMMs) of each of the 120 markers using the bit score gathering threshold (GA) for each profile to define a corresponding homolog.

We proceeded to an additional quality checking as previously[112]. We removed genomes in which more than 90% of markers were detected, and eliminated markers detected in less than 90% of genomes. Multiple sequence alignments (MSAs) were then performed with MAFFT v7.490[113], trimmed with BMGE v1.12[114] for each marker, and concatenated into a single alignment. Trees were inferred using the maximum likelihood method implemented in the IQ-TREE v2.2 software[115,116] and a substitution model selected from the global alignment with ModelFinder[117] with the –MFP option. Branch robustness was estimated by the SH-like approximate likelihood ratio test[118] (1000 replicates) and the ultrafast bootstrap approximation[119] (1000 replicates). Trees were drawn and edited using FigTree v1.4.4[120] and Inkscape software (Inkscape Project, 2020, https://inkscape.org).

Identification of the protist relied on a maximum-likelihood phylogenetic tree of the *Ciliophora* constructed from SSU 18S rRNA gene sequences alone, as no eukaryotic draft genome and other conserved marker genes could be obtained. A sample of 294 sequences at least 1634 pb long representing 205 *Ciliophora* families was retrieved from the PR2 database (https://app.pr2-database.org). The tree was rooted with the *Colponemidia* based on Tikhonenkov et al.[121], who demonstrated their monophyly and position as the sister lineage to all other known *Alveolata*. The alignment of the 319 sequences was performed using MAFFT and trimmed using a relaxed parameterized Gblocks v 0.91 program[122]. The tree was then inferred as described above for whole-genome trees.

## Functional annotation and metabolic network modeling

Metabolic pathways were inferred from the MetaCyc database[66] as described previously[112], which offers an extensive catalog of metabolic pathways, enzymatic reactions, enzymes, chemical compounds, genes, and review-level information for organisms in the three domains of life. This database includes 3105 pathways and 18,566 reactions in version 27.0, which was used in this study. Metabolic pathway reconstruction was performed using the PathoLogic algorithm of Pathway Tools. Beforehand, Prokka v 1.14.5[110] was used to perform gene calling for each genome. Then, functional annotation was performed with KofamScan v.1.3.0[123], which assigns KEGG Orthologs (KO) family numbers to protein sequences using HMMs database of KO and the pre-computed adaptive score thresholds. Next, we created Pathway Tools input files with associated enzymatic activities for protein-coding genes described with either Enzyme Commission (EC) numbers or MetaCyc reaction identifiers when cross-references between KO, KEGG, and MetaCyc identifiers were available. Finally, the obtained Pathway/Genome Databases (PGDB) were queried using the Python-Cyc API v2.0.2 (https://github.com/networkbiolab/PythonCyc) to compute pathway completeness rates for each genome and a pathway/genome matrix. Scripts that were used to generate Pathway Tools input files and the matrix are available at: https://github.com/labgem/metacoco[124].

The genomes were also analyzed using the MicroScope platform[104] for further comparative analysis and expert functional annotation. Protein-coding genes were classified by eggNOG-mapper version 2.1.12[56] and the public database eggnog v 5.0.2[125]. Putative virulence-associated genes were identified using VirulenceFinder[126] and BLASTP searches against hits of the Virulence Factor Database (VFDB)[127], applying a minimal identity of 30%.

## Statistics and reproducibility

All experiments were independently performed at least three times with similar results. Microscopy observations (including optical, confocal, electron, and X-ray–based imaging) were also conducted on at least three independent occasions, yielding consistent qualitative observations across replicates.

## Reporting summary

Further information on research design is available in the Nature Portfolio Reporting Summary linked to this article.

## Data availability

Data generated or analyzed during this study are included in this published article and its Supplementary Information and Description of Additional Supplementary Files. The sequencing data were deposited on public databases. The 16S rRNA gene amplicon sequences have been deposited in the NCBI Genbank database under the accession numbers OR342250-OR342267, OR294250-OR294271, OR250250-OR250289, OR294289-OR294298, and OR294333-OR294370. The genome assemblies were deposited in the NCBI BioProject database under the accession number PRJEB65892.

## Code availability

Scripts are available at: https://github.com/labgem/metacoco [124].

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

## Acknowledgements

This work was supported by the French National Research Agency (SymbioMagnet ANR-21-CE02-0034-01 and ANCESMAG ANR-20-CE92-0050) and the CNRS—mission pour les initiatives transverses et interdisciplinaires (MITI), adaptation du vivant à son environnement (*SymbioAdapt* project). R. Bolzoni's PhD contract was supported by the CNRS—MITI. This work benefited from access to ALBA and has been supported by iNEXT-Discovery, project number 2022025566, funded by the Horizon 2020 program of the European Commission. Synchrotron beamtime at ID16B of the European Synchrotron Radiation Facility (ESRF) is also acknowledged under proposal LS2983. We acknowledge the Institut de Radioprotection et de Sûreté Nucléaire (IRSN) at CEA Cadarache for the access to the transmission electron microscope Tecnai G² BioTWIN. This work received support from the French government under the France 2030 investment plan, as part of the Initiative d'Excellence d'Aix-Marseille Université—A*MIDEX, and is part of the Institute of Microbiology, Bioenergies and Biotechnology - IM2B (AMX-19-IET-006). We are grateful to the INRA MIGALE bioinformatics platform (http://migale.jouy.inrae.fr) for providing computational resources. The LABGeM (CEA/Genoscope & CNRS UMR8030), the France Génomique and French Bioinformatics Institute national infrastructures (funded as part of Investissement d'Avenir program managed by Agence Nationale pour la Recherche, contracts ANR-10-INBS-09, ANR-11-INBS-0013 and ANR-21-ESRE-0048) are acknowledged for support within the MicroScope annotation platform. F. Abreu acknowledges the Brazilian agencies CNPq, FAPERJ, CAPES, the microscopy facility CENABIO, and Dr. Jefferson Cypriano for their support in TEM observation. We thank Zoé Rouy for data submission to public databases. We thank Matthieu Amor, Tanguy Le Borgne, and Camille Bouchez for their help in sampling the aquatic environments in Britany.

## Author contributions

C.L.M., D.V., K.B., and C.T.L. designed the project, analyzed and interpreted the data. C.L.M., R.B., B.A., D.M.C., C.G., N.M., M.B., S.F., V.dC., F.S-P., E.P., A.D., D.V., C.C., F.A., K.B., and C.T.L. performed the experiments and interpreted the data. C.L.M. and C.T.L. prepared the manuscript.

## Competing interests

The authors declare no competing financial interests.
