## [Transparent Peer Review File · Nature Communications]

Magnetoreception in a freshwater ciliate arises from endosymbiosis

Corresponding Author: Dr Christopher Lefevre

Version 0:

Reviewer comments:

Reviewer #1

(Remarks to the Author)

Review of Bolzoni, Monteil et al. (NCOMMS-25-28667-T)

Bolzoni, Monteil et al. describe the discovery of a novel magnetosymbiosis between a freshwater ciliate and its prokaryotic endosymbionts. The ciliate, belonging to a novel clade within Prostomatea, is magnetotactic (north-seeking) thanks to magnetite minerals produced by its symbiont (belonging to the Desulfovibrionaceae). This symbiosis is reminiscent of the magnetosymbiosis between a distantly related euglenozoan and a magnetotactic Desulfobacteriales ectobacterium, with the key difference that here the symbiont appears to have been adapted to intracellular lifestyle. The ciliate additionally hosts 3 other non-magnetotactic endosymbionts (Desulfoarculaceae, Midichloriaceae and Coxiellaceae) as well as diverse diatoms, whose role for the host is not clear.

The authors' discovery not only describes a wholly novel intriguing symbiosis but it further reveals the profound effect symbiotic interactions have on eukaryotic evolution.

I very much enjoyed reading this manuscript; I find the topic of protist symbioses fascinating, the manuscript is very well written, and the images and videos are spectacular. The study encompasses a truly impressive collection of data to support the authors' conclusions, including advanced microscopy (SEM, TEM, nano-XRF) techniques and metagenomics. My comments largely pertain to minor clarifications in text (see below) but I have one general question:

1. The authors don't explicitly state whether the ciliate host is anaerobic or not. As it lives below the oxic-anoxic transition zone, is 'sensitive to oxygen' and contains putative hydrogenosomes it might likely be an anaerobe. On the other hand, phylogenetically, it belongs to a sister clade of the mainly aerobic Prostomatea. Do the authors lean one way or another? Did they maybe find indications in their molecular data (e.g. potential mitochondrial proteins etc.) that would allow them to address this point?

This is not just a curiosity issue – the metabolism of the host is a crucial consideration with regard to the metabolic interplay with its many endosymbionts. Two of these are obligate anaerobes, the Coxiellaceae is presumably an aerobe (is it?), and then there is the photosynthetic diatom and maybe even microalgae. How can these all sensibly work together? I'd very much like to see this addressed with a few sentences in the manuscript.

Line-specific comments:

Line 55: It is not my impression that the importance of symbiotic interactions has been 'long-denigrated'. Is this with regard to a specific reference? I'd suggest removing this.

Line 65: replace 'within the host cytoplasm' with 'within the host cell' to encompass also intranuclear bacteria.

Line 73-79: Please include corresponding references. Also, given the recent re-organization in taxonomy, it might be worthwhile mentioning that Pseudomonatota were previously referred to as proteobacteria.

Line 105+118+275: replace 'degenerated' with 'reduced'

Line 126-127: it is not clear to me how magnetotaxis alone would allow organisms to 'move efficiently towards their optimal redox niche'. To find redox conditions (e.g. oxygen) the organisms would need to be additionally aerotactic or chemotactic.

Line 165ff: the link between the arguments 1 and 2 and the conclusions that the host is 'fragile to osmotic changes and sensitive to oxygen' is not clear to me. Both argument 1 – that less ciliates were enriched using a stirring bar and argument 2- that the cells survived longer in a hanging drop than on a glass slide would suggest that they are sensitive to mechanical force (and maybe heat). I don't see how the conclusion about sensitivity to osmotic change and particularly oxygen was made. The oxygen tolerance is an important issue and evidence regarding this should be clearly outlined.

Line 191: this statement is in contrast to the one made later (Line 622) where it is suggested that the ciliate can take up whole intact diatoms. Please remove or revise.

Line 209-211: It is not clear to which organisms/clades the different identity values refer to.

Line 212-216: I might have missed the corresponding method section that describes how the 18S phylogeny was calculated and which sequences were included in the tree. If indeed missing, this information needs to be added to the manuscript. Additionally, the 18S tree in Suppl. Fig. 4 is hard to read, as bootstrap values and names overlap. This should be checked and amended.

Line 224: 'exposure to oxygen' – do you mean air?

Line 237: delete 'as well' as it is repetitive with 'also'.

Line 264-265: What it meant with '...the (bacteria) showed no connection to the extracellular environment...' ? Please clarify and revise.

Line 323: should read 'burnetii'

Line 326: unclear what is meant with 'the same ecological niche', please clarify.

Line 336-337: It is not obvious to me that the Desulfovibrionales in the CLSM picture is at the anterior part of the cell, as mentioned in text. Rather it seems to stretch across the whole length of the host. As the FISH image in Fig. 6a and in Video 8 are the same cell, the authors should include a few more representative FISH images to support the claims about symbionts' location and abundance.

Line 343-345: From this one FISH picture it seems to me that the gammas are in fact more abundant than the Desulfovibrionales, in contrast to what is stated in text.

Line 351: what is meant with 'symbiotic counterparts', who is this referring to?

Line 354: I find a protein-coding density of 57% very low and rather atypical for an endosymbiont – the authors could touch on this and the high pseudogene presence in their discussion.

Line 358: 'typical of obligate endosymbionts' is vague and unclear what this refers to. Be more concrete or remove such generic statement.

Line 360: please include how many % of all proteins these three categories amounted to.

Line 366: 'apart from Dsd-1, a last category outcompeted all the others:...' – what do you mean with this? Please revise and split these two statements into two sentences for better readability.

Line 372: Who are the Pseudomonadota symbionts? In the diversity description Pseudomonadota are not mentioned.

Line 382: Please use the taxonomic names for each symbiont consistently throughout the text and do not switch between e.g. GD-1, Legionellales, Gammaproteobacteria when describing the same organism as this complicates the reading.

Line 406: do you mean oxidation of organic compounds (such as acetate)? Or really degradation (of polymers, such as PHB or such)?

Line 413: do you mean it was not complete (rather than not fully predicted) in the genome?

Line 441ff: I don't understand this sentence. This study does not report the euglenozoan holobiont. Please revise.

Line 446: I'd recommend to remove the 'five-member' as it is not clear if other symbionts other than the Dsd-1 are necessary for functioning of the consortium and also 5 member reference omits the diatom.

Line 458: 'Unfortunately, ...' What is meant with this sentence? It does not make sense here, I suggest removing.

Line 484: Please include the number.

Line 496: 'should be largely enough'. As you document magnetotactic behavior of the ciliate, it is clearly enough. Please revise accordingly.

Line 503: reminding how? Please amend the statement to be more concrete.

Line 528: I am not sure I follow why the discovery in France and Brazil would raise question about the presence of magnetite biomineralization in microeukaryotes. Do you maybe mean biogeography or distribution or origin? Please be more specific.

Line 530: Revise this formulation – I guess you mean association of protist host from different lineages with symbionts from different phyla?

Line 532-533: It should also be considered here that the emergence of these symbioses will be likely facilitated in environments where magnetotactic prokaryotes are prevalent and common.

Line 539: should be 'consuming' not 'producing'

Line 542 - 547: which other hydrogen-producing specialized organelles are meant here? The protist can utilize 'external inorganic electron acceptors'? Which would these be? 'Offload' sounds a bit like slang. Also, hydrogen doesn't need to be 'offloaded', it will diffuse out of a swimming ciliate without problems. What about the sulfide that is the end product of the Desulfobacterota metabolism – how does the host deal with that toxic substance being produced intracellularly?

Line 567: I would not say that Rickettsiales are better known for mutualism as many people (including me) might associate them with human pathogens. Or do you mean the specific group that includes your symbiont?

Line 570: What is MHB? I missed this explanation.

Line 571: Spell out whose genome.

Line 575: Which associations and lost from where, which genomes/organisms?

Line 582: I would not describe being prey as a facultative relationship – or what is meant here?

Line 590-591: Is this the case in this host? Could you include a reference to a Figure then?

Line 598 – 599: Also, I assume that sediments will not be limited in any of these nutrients, unlike the water column.

Line 599ff: I wonder - if the diatom (or its chloroplasts) are used by the host for photosynthesis what happens with the produced oxygen? Can you speculate how the obligately anaerobic sulfate reducers can possibly deal with this?

Line 652: Care should be taken to clarify that in the euglenozoa symbiosis magnetotaxis of the ectosymbiont is combined with the chemotaxis (aerotaxis) of the host, and thus allow navigation in chemically-stratified aquatic environments. Is this ciliate host also capable of aero- or chemo-taxis?

Line 657: This is to me an unclear statement and it is a pity to end up this really nice paper on this kind of vague sentence. Maybe try to revise to be more specific.

Paragraph Line 675ff: What is meant by microcosms? Please explain. It is unclear to me how the oxygen sensor was calibrated – what is 'saturated humid air (O₂ saturation 100%)'? Do you mean pure oxygen? What is humid? Why was the

calibration not done with air-saturated water, as outlined in the instruction manual? What is 'water solution flushed with N2'? Is it MilliQ? Was it flushed or bubbled? How long? Please include all relevant information here.

Paragraph Line 788ff: It is essential to include here the formamide concentrations of the hybridization buffers, which were used for FISH.

Line 1260: replace 'blows' with 'breaks apart'

Line 1279: replace 'presence of the extremity of a diatom' with 'part of a diatom'.

Line 1325: replace 'authentication' with 'identification' or 'visualization'

Line 1328: replace 'authenticate' with 'target'

Line 1337ff: Please revise this sentence for clarity. Also, sulfate reducers typically do not use more complex organic compounds.

Line 1340-1341: replace with '... and might depend on the biosynthetic capacities of other symbionts for their growth'.

Reviewer #2

(Remarks to the Author)

The paper entitled "Magnetoreception in a freshwater anaerobic ciliate arises from endosymbiosis" represents the first documented case of magnetoreception in a protist resulting from an endosymbiotic association with bacteria. While magnetoreception in protists remains a rare and poorly understood phenomenon - previously observed only in protists with ectosymbionts, this study reveals an astonishingly complex and novel system. The discovery not only expands our understanding of magnetoreception in eukaryotes, but also contributes to the growing body of research highlighting the remarkable diversity and evolutionary significance of interactions between bacteria and protists.

As the first discovery of a system in which magnetoreception is acquired through endosymbiosis, this research is both novel and timely and fits well into the current wave of studies investigating endosymbiotic interactions between bacteria and protists. The results are thoroughly documented and supported by multiple lines of evidence, including advanced microscopy techniques and comprehensive genome analyses.

However, there are concerns regarding the interpretation of the results. While the involvement of the endosymbiont DsD-1, a member of the Desulfovibrionaceae, in magnetoreception is well supported, the specific contributions of other members of the holobiont, both bacterial and eukaryotic, remain unclear. These additional partners may play an important role in supporting the anaerobic lifestyle and overall fitness of the host, but their direct involvement in magnetoreception has not been convincingly demonstrated. Therefore, characterising the phenomenon as "magnetotaxis of a multi-partner endosymbiotic interaction" appears premature based on current data and their interpretation.

The manuscript repeatedly emphasises that magnetoreception arises from the interaction between several partners; however, the evidence presented does not sufficiently support this claim. Either more direct evidence is needed to demonstrate the involvement of all partners in magnetoreception, or the current formulation should be weakened to more accurately reflect the limitations of the data. Ideally, experiments on cultures would clarify the critical role of individual partners, although this is often difficult to achieve. Therefore, a more detailed reconstruction of metabolism that emphasises the interdependence of all holobiont components is required, together with more careful interpretation.

Detailed comments on these and other issues are provided below. Overall, the study presents a convincing result that is supported by solid data. However, our main concerns relate to the way in which the results are communicated. In some cases, the claims are presented too broadly, and in others, the interpretations appear too speculative. We believe that a more focused presentation on the magnetoreception aspect would strengthen the manuscript.

Detailed comments

Line 42 – "However, in this case, magnetotaxis arises from a multi-partner endosymbiotic interaction." Is the magnetoreception in the case presented strictly dependent on multiple endosymbionts (instead of just single magnetotactic symbiont), in contrast to previously known magnetoreception in euglenozoans (relying on extracellular bacteria)?

Line 65 – There are known cases of endosymbionts residing in the nucleus (which formally is distinguished from the cytoplasm) (10.1016/j.tcb.2015.01.002), thus it is better to change for the "integrated within the host protoplasm" or "integrated within the host cytoplasm and/or nucleus".

Line 104 - please change protozoa to protist

Line 105 - please use mitochondria-related organelles (MRO) as a well-established term for reduced mitochondria in anaerobic lineages of protists

Line 110 – rather "protection against competitors or predators", not "chemical camouflage" (the latter not mentioned in the original publication).

Line 204 - In eukaryotes, genetic distance based on 18S rDNA is generally not sufficient to make reliable taxonomic decisions, as evolutionary rates vary greatly between lineages. The guidelines of the Zoological Code should be followed when describing new ciliate species. The data presented may not be sufficient to formally describe a new species or genus, as this requires the identification of type material. Furthermore, a formal taxonomic description does not appear to be necessary in this case, and its absence would not diminish the significance of the results. We therefore suggest reconsidering whether such a description is needed.

Line 287 - we suggest providing the information on the average genome coverage for the obtained bacterial genomes. This will better illustrate the quality of data presented and may inform about relative abundances of the symbionts in the host cell.

Line 288 - how the estimation of the amount of eukaryotic reads was performed? As the authors attempted total rRNA extraction from the raw assembly, is it correct to state that no 18S fragments of either host or intracellular photosynthetic algae were recovered? If so, please mention this in an appropriate section of the results.

Lines 298-299 - as Dsd-1 is currently the only endosymbiotic magnetotactic bacterium in its family, authors may consider comparing its genomic capabilities with that of the free-living relatives (*Oceanidesulfobivrio* and *Megalodesulfobivrio*). Similarly, comparison with *Ca. Desulfarcum epimagneticum* CR-1 may inform about the differences in the integration between eukaryotic hosts and their magnetotactic symbionts.

Lines 339-341 - Presence of the mam genes is mentioned in support of the ability of Dsd-1 to form magnetite particles. While this proof seems sufficient, Extended data Fig. 8e shows that some of the core mam genes (according to Lefevre et al. 2013 <https://doi.org/10.1111/1462-2920.12128>) are missing (mamA, mamE, mamQ, mamB). Was there an attempt to find those genes in the assembly in order to confirm or rule out that their absence was caused by binning inaccuracy? Additionally, while unnecessary here, the phylogenetic analysis of mam genes would be instrumental in understanding the process of biomineralisation cluster acquisition.

Line 354 - How were the pseudogenes identified? This should be included in the methods.

Lines 364-366 – we find the information about the proportion of proteins related to glycosyltransferases a bit out of context, as it is not interpreted or mentioned further. Is this proportion different in other symbionts or somehow remarkable in this case?

Lines 366-368 - there is an enormous number of replication, recombination & repair genes identified in genome Dcd-1 (as shown in Supplementary Table 1), an order of magnitude larger than in other symbionts or free-living bacteria used for comparison. Is it possible to assess whether this inflated number is not caused by the mobile genetic elements-related genes? They often fall within the same COG category and may contribute to the assembly fragmentation. In addition, their identification informs about the proliferation of mobile genetic elements, observed in some symbiont genomes.

Lines 368-371 - the loss of the replication, recombination & repair machinery in Dsd-1 is very interesting, as those processes are highly conserved and mostly retained even in extremely reduced endosymbionts with genomes below 1 Mbp (10.1146/annurev-micro-091213-112901 and 10.1371/journal.pbio.3002577). Therefore, we suggest presenting more information on the components that are preserved or lost, and whether this loss would impact the bacterial capability to maintain its genome. It is important information for understanding the genome reduction processes in endosymbionts.

Line 371 - We suggest adding a reference to the Supplementary table 1 as well, because Fig. 5e shows the relative frequencies of gene categories and not the absolute number of genes, which is important for interpreting the gene losses of conserved cellular systems.

Line 372 - Proteins with typical eukaryotic-interacting domains (including leucine-rich repeats) are presumed to participate in symbiont manipulation of the host cell after delivery via secretion systems (10.1016/j.cub.2021.05.049). While it is possible that they were acquired via HGT, there is no direct evidence for that presented, and we would suggest focusing on their potential role in interaction with the host. In particular, we suggest providing information regarding the number of such genes for a particular symbiont genome.

Line 387 - We suggest moving the details about the virulence factor search into the methods section.

Line 395-397 - We suggest toning down this interpretation of function as many of the presented virulence factor predictions are based on relatively low (30%) identity hits and may indicate only distant similarity and not the same function.

Line 439 - We suggest rephrasing this, as there is no evidence for reliance of magnetoreception specifically on syntrophy, to the best of our knowledge. E.g. there is a possibility that those layers of interaction are uncoupled, or that magnetoreception was a driving force for the evolution of this association, with established syntrophy being a later byproduct.

Lines 474-476 - we strongly encourage authors to provide full, proper descriptions of the newly described bacteria (see examples, 10.1038/s41396-023-01499-6 and 10.1038/s41467-024-54047-x). Authors may also consider using SeqCode (10.1038/s41564-022-01214-9) for new taxa submission.

Lines 501-504 - presented statement suggests that the magnetite-forming endosymbiont has lost many of its metabolic

capabilities, preserving only essential pathways for cell functioning. However, it still possesses a relatively large genome (2 Mbp) and in terms of the number of metabolic pathways identified is only the second lowest after the Midichloriaceae endosymbiont. Additionally, it is suggested to participate in syntrophy. We suggest to tone this down a bit or provide more details about the relevant pathway losses. For example, severe deficiencies of systems important for cell maintenance (phospholipid/fatty acids synthesis, peptidoglycan/cell wall synthesis, division, nucleotide and amino acid synthesis) may support this claim.

Line 569-570 - did authors consider performing a search for the ATP/ADP translocases in endosymbiont genomes? This information may be useful for assessing the potential impact of Pseudomonadota symbionts on the energy metabolism of the holobiont.

Line 571-573 - we find this part slightly confusing, are those auxotrophies present in Pseudomonadota symbionts? If yes, how do they contribute to the consortium? Are there any contributions from the Desulfobacterota symbionts? Figure 6b uses the generalised term "amino acids" and only provides information about vitamins B1 (provided by all symbionts), B9 (provided by three symbionts) and B12 (provided by two symbionts), impairing interpretation.

Line 575-576 - Families of Rickettsiales are considered to be relatively old groups of intracellular bacteria, but associations with particular hosts may be more recent (10.1038/s41467-021-23645-4, 10.1038/s41467-024-45351-7 and 10.1016/j.cub.2021.05.049). On the other hand, UBA9339 symbiont resides in a poorly explored group of orders related to Legionellales and Coxiellales. While there are suggestions that host association in general may be ancestral in these lineages, even its obligatory nature may be a more recent trait (10.1038/s41564-022-01174-0). We suggest explaining this notion in more detail or citing a specific reference.

Line 577 - The role of diatoms in this system remains highly speculative. Ciliates often form photosymbioses with green algae, although they have never acquired permanent plastids. Such interactions have not yet been observed between ciliates and diatoms, and although the possibility is intriguing, convincing evidence is lacking. We therefore recommend focussing on the most important discoveries and not on the uncertain role of diatoms in this context.

Lines 589-590 and 621-623 - Previously (lines 190-191), authors suggest that examined magnetotactic ciliates are unable to phagocytose prey larger than small bacteria due to the small cytostome size, contradicting themselves.

Lines 623-624 - This sentence is confusing, as it seems to conflict with the prior one.

Lines 624-625 - citation needed.

Lines 602-603 - wouldn't the immediate proximity of oxygen producers be highly detrimental for all anaerobic members of the consortium?

Lines 652-658 - the link between particular magnetotactic symbiosis and eukaryogenesis suggested here is unclear.

Reviewer #3

(Remarks to the Author)

The manuscript by Bolzoni et al. presents the identification a new class of magnetotactic eukaryotes distinguished by an symbiotic relationship between a magnetotactic bacterium and a protist. Previous work, including groundbreaking studies from this group, had shown that protists can be magnetically sensitive in two different ways: passively through the ingestion of magnetotactic bacteria and actively through an ectosymbiotic relationship with a magnetotactic bacterium. This work presents the incredible discovery of an endosymbiotic magnetotactic bacterium. The cell biology, multi-layered endosymbiotic relationships, mineralogy, phylogeny, and genomics of this new group of protists is detailed through a series of carefully performed experiments. The findings are highly impactful and open up a host of new questions for microbiologists, cell biologists, and evolutionary biologists. I am thoroughly impressed by the work and cannot wait to see it published. I have only a few minor comments which can be addressed through changes to the text and figures.

1. Line 247: There is no reference to a figure but there is a closed parens.

2. Line 253: There is a reference to "membrane similar to bacterial magnetosomes" in ED Figure 6d. First, I had to zoom in quite a bit to see the relevant magnetosomes. To my eyes these do not look as convincingly like the ones in the Gorbey reference. Given that there is some dispute over the presence of membranes around bullet-shaped magnetosomes, I would encourage the authors to provide a more convincing image or acknowledge the possible ambiguity.

3. Lines 339-341: I was surprised at how little attention and space was given to analyzing the magnetosome genes of the endosymbiont. This is incredibly important in thinking through the evolutionary diveristy of magnetotactic bacteria and should be given a more visibility. I would encourage the authors to have a dedicated figure for these genes (extended data is fine although it could also work as a portion of a main text figure). The current extended data figure (8E) has some ambiguous descriptions in its legend. For instance, line 1425 says that each arrow represents a gene corresponding to MSR-1 operons. However, MSR-1 does not have mad genes. In the legend, "mam" should be italicized and CR-1 should be defined in the legend. I would request a table comparing the magnetosome proteins of Dsd-1 with their accession numbers to those of other magnetotactic bacteria (for instance, including the % identity to homolog in RS-1 or others). This would be a

big contribution to the community.

4. Finally, I feel that the extrapolation from genome sequences to metabolic capability of the endosymbionts was too strong. The authors used the words "can" and "cannot" based on the presence or absence of certain genes. However, one cannot say that a certain reaction will happen because genes are there as some might be inactive. Similarly, novel pathways using genes annotated as hypothetical or as having different function may fulfill the missing pieces in a metabolic pathway. In other sentences, the authors used "may" to describe the potential activities of these organisms. This is more appropriate and should be repeated throughout.

5. Finally, I could not open videos 1, 2, and 5. I am not sure if it is an issue with my current setup or something is wrong with the files.

Reviewer #4

(Remarks to the Author)

Version 1:

Reviewer comments:

Reviewer #1

(Remarks to the Author)

I am satisfied with the revisions the authors made to the manuscript and I appreciate the time and effort they took to address all reviewers' comments. In my opinion, the revision has improved the manuscript and I am looking forward to seeing this wonderful discovery published.

I have only a few last small suggestions for text edits:

Line 56: to me 'diversification' is redundant here and could be deleted

Line 75-76: It is my impression that while the endosymbiotic origin of plastids and mitochondria is largely undisputed, discussions regarding the origin of nucleus persist (i.e. the autogenic models). Hence, I'd suggest to rephrase it here as such (e.g. : '... emergence of plastids, mitochondria, and potentially also the nucleus').

Line 76: Consider replacing 'most' with 'many'.

Line 78: mention also aerobic respiration, as the primary role of mitochondria

Line 140: 'multi-partner' instead of 'multi-partners'

Line 250: replace 'particles' with 'which'

Line 254: consider adding 'magnetite' before 'particles' for clarity.

Line 262 – 264: consider rephrasing: 'All bacteria were considered endosymbionts as they appeared to be completely enclosed ... and the extracellular space.'

Line 272: replace 'support' with 'confirm'

Line 280: replace 'genetic' with 'taxonomic'

Line 299: replace 'population belongs' with 'populations belong'

Line 356: As it is only 4 genomes, please add also the sizes of the two other genomes (DsD and GD-1).

Line 363: replace 'proteins-coding' with 'protein-coding'

Line 378: please italicize gene names.

Line 382: what about DsD? Please add.

Line 421: replace 'account' with 'accounts'

Line 428 – 431: sentence is unclear. Does 'exclusively' mean that only menaquinone and no other quinone was detected in DsD or that menaquinone was exclusively detected in DsD and in no other symbiont? Please clarify. Also, it is not obvious to me how can the quinones of one endosymbiont have such broad implications for the whole consortium?

Line 433: Do you mean 'or' instead of 'of'?

Line 453: I suggest to slightly rephrase, e.g.: 'A recent discovery of marine magnetic holobionts revealed a unique ability of certain microeukaryotes ...'

Line 472: replace 'with' with 'from'

Line 473 – 474: replace with '...not possible as the genome of the host could not be assembled ...'

Line 562-565: 'On the other hand, DcD-1 and DsD-1 could enable their host to breathe sulphate instead of oxygen.'

Please rephrase. Indeed, DcD-1 and DsD-1 can breathe sulphate, and thus so can the holobiont but not the host (= ciliate) as there is no evidence that the ciliate can access the energy obtained from sulfate reduction, unlike in ref. 81. The authors themselves proposed that the ciliate obtains energy from fermentation in the hydrogenosomes. This should be clarified.

Line 653: replace 'highlights' with 'reports the discovery of' and replace 'capable to sense' with 'capable of sensing'

Figure Legend:

Fig. 1b: In the text white and grey arrows are mentioned but in the image there are only black arrows. Can you check and correct if necessary?

Fig. 4a: I can only see 2 arrows (=2 cells with magnetosomes), whereas 4 are mentioned in the figure legend.

Fig. 5a,b,c,d: the symbionts from this study should be in bold in the respective trees but they are not (in my version of the figures). Please check and make sure to highlight them (in bold or colour).

Line 1356: Replace 'under study' with 'from this study'.

Fig. 6b: replace 'preys' with 'prey' in sketch

Line 1377: replace 'cell host' with 'host cell'

Line 1390 - 1392: These two sentences are essentially identical, please delete one.

Reviewer #2

(Remarks to the Author)

The authors have carefully addressed our comments, as well as those of the other reviewers, and have made substantial improvements to the manuscript. All suggested changes were thoroughly considered and implemented where possible, either through additional analyses or by removing debatable statements. The main story is now much more clearly articulated and supported by additional data, while more speculative aspects have been either removed or appropriately rephrased.

Some of our comments proposing additional analyses, such as phylogenetic analysis of mam genes or more in-depth comparative analyses, were intended as general suggestions. While these were not fully pursued, we understand and agree with the authors' decision, as incorporating them would have substantially expanded the scope of the manuscript and blurred the main focus.

We have no further comments, and we would like to congratulate the Authors on this exciting and well-supported discovery in protist symbiosis and look forward to seeing the paper published.

Reviewer #3

(Remarks to the Author)

The authors have answered all of my concerns. Congratulations on a fantastic manuscript.

Reviewer #4

(Remarks to the Author)

Manuscript number: **NCOMMS-25-28667-T**

Title: **Magnetoreception in a freshwater ciliate arises from endosymbiosis**

Response: We would like to express our gratitude to the editor for giving us the opportunity to address the concerns raised by the reviewers in this new submission. We have carefully considered their feedback and found the overall comments and suggestions constructive. We believe we have addressed all the requested changes.

In this revised manuscript, which we believe has been significantly improved, we have provided a point-by-point response to each concern. We have included line by line where changes were made in the revised version.

Reviewers' comments/questions are written in black while our answers are written in blue.

See below.

Reviewer's Comments:

Reviewer #1 (Remarks to the Author)

Review of Bolzoni, Monteil et al. (NCOMMS-25-28667-T)

Bolzoni, Monteil et al. describe the discovery of a novel magnetosymbiosis between a freshwater ciliate and its prokaryotic endosymbionts. The ciliate, belonging to a novel clade within Prostomatea, is magnetotactic (north-seeking) thanks to magnetite minerals produced by its symbiont (belonging to the Desulfococcaceae). This symbiosis is reminiscent of the magnetosymbiosis between a distantly related euglenozoan and a magnetotactic Desulfobacterales ectobacterium, with the key difference that here the symbiont appears to have been adapted to intracellular lifestyle. The ciliate additionally hosts 3 other non-magnetotactic endosymbionts (Desulfoarcuaceae, Midichloriaceae and Coxiellaceae) as well as diverse diatoms, whose role for the host is not clear. The authors' discovery not only describes a wholly novel intriguing symbiosis but it further reveals the profound effect symbiotic interactions have on eukaryotic evolution.

I very much enjoyed reading this manuscript; I find the topic of protist symbioses fascinating, the manuscript is very well written, and the images and videos are spectacular. The study encompasses a truly impressive collection of data to support the authors' conclusions, including advanced microscopy (SEM, TEM, nano-XRF) techniques and metagenomics. My comments largely pertain to minor clarifications in text (see below) but I have one general question:

Response: We thank the reviewer for their kind comments.

1. The authors don't explicitly state whether the ciliate host is anaerobic or not. As it lives below the oxic-anoxic transition zone, is 'sensitive to oxygen' and contains putative hydrogenosomes it might likely be an anaerobe. On the other hand, phylogenetically, it belongs to a sister clade of the mainly aerobic Prostomatea. Do the authors lean one way or another? Did they maybe find indications in their molecular data (e.g. potential mitochondrial proteins etc.) that would allow them to address this point?

This is not just a curiosity issue – the metabolism of the host is a crucial consideration with regard to the metabolic interplay with its many endosymbionts. Two of these are obligate anaerobes, the Coxiellaceae is presumably an aerobe (is it?), and then there is the photosynthetic diatom and maybe even microalgae. How can these all sensibly work together? I'd very much like to see this addressed with a few sentences in the manuscript.

Response: Indeed, it is difficult to make strong conclusions on whether the ciliate is strictly anaerobic or not. With the genome of the ciliate it would have been possible to make stronger conclusions. In this study, we unfortunately did not obtain any usable bins for the analysis of the ciliate genome, our single-cell sequencing only allowed the assembly of the bacterial genomes.

We agree with this reviewer that it is confusing to see such interactions below the oxic-anoxic transition zone. Our observations do not allow us to determine whether the microalgae within the ciliate are alive, nor whether they were phagocytosed by the ciliate as a result of their sedimentation in the sediment. It is also possible that the ciliate is a facultative anaerobe, although our observations indicate that they are consistently found in oxygen-depleted sediments.

In the discussion we addressed the comment of this reviewer with the addition of the following sentences **line 544, page 12**: *“The emergence of these symbioses is likely facilitated in environments where magnetotactic prokaryotes are prevalent and common. This suggests that environmental pressures in oxygen-deprived aquatic habitats might commonly favour the development of such symbiotic relationships among microeukaryotes. However, the strict anaerobic lifestyle of the ciliate host remains under question. Indeed, although it lives below the oxic-anoxic interface, is sensitive to air, contains putative hydrogenomes and has established symbiosis with anaerobic bacteria, we are missing genomic information that would confirm this point. Moreover, the presence of a microalgae questions on how these can all sensibly work together”*.

Importantly, no cytochrome c oxidases were detected in the gammaproteobacterium and the alphaprotobacterium genomes (see KOfamScan outputs in **Supplementary Table 9** in the revised manuscript). If such absence is not related to the draft assembly status, it means that they are not aerobic, but could, as the protist, rely on the sulphate reducing symbionts to offload reduced products (as discussed **line 432, page 10**).

Based on this comment, and after re-reading the article and preparing a revised version, we believe that it would be more appropriate to remove the term “anaerobic” from the title of the article. Accordingly, we have modified the title of the manuscript to: *“Magnetoreception in a freshwater ciliate arises from endosymbiosis.”*

Line-specific comments:

Line 55: It is not my impression that the importance of symbiotic interactions has been ‘long-denigrated’. Is this with regard to a specific reference? I'd suggest removing this.

Response: It is in fact due to an old assumption before Margulis' work, we removed this notion that was making the sentence more difficult to read.

Line 65: replace ‘within the host cytoplasm’ with ‘within the host cell’ to encompass also intranuclear bacteria.

Response: The correction has been made (**line 64, page 3**).

Line 73-79: Please include corresponding references. Also, given the recent re-organization in

taxonomy, it might be worthwhile mentioning that Pseudomonatota were previously referred to as proteobacteria.

Response: The following reference was added at the end of this sentence: López-García, P. & Moreira, D. The symbiotic origin of the eukaryotic cell. *C R Biol* **346**, 55–73 (2023).

Line 73, page 3, in parentheses we mentioned that *Pseudomonatota* was “previously referred as *Proteobacteria*”.

Line 105+118+275: replace ‘degenerated’ with ‘reduced’

Response: The correction has been made.

Line 126-127: it is not clear to me how magnetotaxis alone would allow organisms to ‘move efficiently towards their optimal redox niche’. To find redox conditions (e.g. oxygen) the organisms would need to be additionally aerotactic or chemotactic.

Response: Indeed, magnetic orientation alone does not allow magnetotactic microorganisms to find their optimal redox niche, magnetotaxis is defined as a combination of the magnetic alignment of the cell while they swim and chemotaxis. We understand the confusion while reading the sentence lines 126-127, we have modified this sentence, **line 125, page 4**, by: “whereby organisms move efficiently towards their optimal redox niche combining guidance by the Earth’s magnetic field lines and chemotaxis”.

Line 165ff: the link between the arguments 1 and 2 and the conclusions that the host is ‘fragile to osmotic changes and sensitive to oxygen’ is not clear to me. Both argument 1 – that less ciliates were enriched using a stirring bar and argument 2- that the cells survived longer in a hanging drop than on a glass slide would suggest that they are sensitive to mechanical force (and maybe heat). I don’t see how the conclusion about sensitivity to osmotic change and particularly oxygen was made. The oxygen tolerance is an important issue and evidence regarding this should be clearly outlined.

Response: We agree with this comment, **line 165, page 5**, we modified the sentence by: “The cells appeared sensitive to mechanical force”.

Line 191: this statement is in contrast to the one made later (Line 622) where it is suggested that the ciliate can take up whole intact diatoms. Please remove or revise.

Response: Line 622, now **line 626, page 14**, of the revised version, we modified the sentence to avoid this confusion: “Because diatoms and protists are similar in size and given the small diameter of the cytostome opening (0.5 μm), it is unlikely that the cytostome can expand enough to engulf the entire rigid silica frustule of diatoms, as also observed in other ciliates such as *Didinium nasutum*⁸⁸. Moreover, the silica composition of the diatom’s frustule may hinder flexibility. Alternatively, diatoms and other algae may be phagocytosed through the plasma membrane of the ciliate and subsequently internalized within phagosomes⁸⁹.”

Line 209-211: It is not clear to which organisms/clades the different identity values refer to.

Response: We clarified the sentence in the revised version of the manuscript, **line 208, page 6**, as follows: “*Although average pairwise identities values between members of Prostomatea and Plagiopylea were approximately 90% and 85% respectively, they were on average 86% and 82% identical with those of the magnetotactic protists, respectively.*”

Line 212-216: I might have missed the corresponding method section that describes how the

18S phylogeny was calculated and which sequences were included in the tree. If indeed missing, this information needs to be added to the manuscript. Additionally, the 18S tree in Suppl. Fig. 4 is hard to read, as bootstrap values and names overlap. This should be checked and amended.

Response: The method for the phylogenetic reconstruction based on the 18S rRNA gene sequences was included in the Figure 1d caption in the previous version but was indeed missing in the materials and methods section. This information is now included in the section “Genome-based taxonomic classification and molecular phylogeny” **line 898, page 19**: “Identification of the protist relied on a maximum-likelihood phylogenetic tree of the Ciliophora constructed from SSU 18S rRNA gene sequences alone, as no eukaryotic draft genome and other conserved marker genes could be obtained. A sample of 294 sequences at least 1634 pb long representing 205 Ciliophora families were retrieved from the PR² database (<https://app.pr2-database.org>). The tree was rooted with the Colponemidia based on Tikhonenkov et al.¹²⁰, who demonstrated their monophyly and position as the sister lineage to all other known Alveolata. A total of. The alignment of the 319 sequences was performed using MAFFT and trimmed using a relaxed parameterized Gblocks v 0.91 program¹²¹. The tree was then inferred as described above for whole genome trees”. The Figure 1d caption was also simplified in the revised version.

Supplementary Fig. S4 has been improved in the revised version of the manuscript. We also included a new supplementary data (now Supplementary Data S1) that corresponds to the newick format of the tree in which all the bootstraps value are visible.

Line 224: ‘exposure to oxygen’ – do you mean air?

Response: We replaced “oxygen” by “air”.

Line 237: delete ‘as well’ as it is repetitive with ‘also’.

Response: The correction has been made.

Line 264-265: What it meant with ‘...the (bacteria) showed no connection to the extracellular environment...’? Please clarify and revise.

Response: To be defined as endosymbiont, a bacterium has to be completely enclosed within the host in the cytoplasm. If a symbiont is physically encapsulated within its host but is in contact with to the extracellular media, it will be considered as an ectosymbiotic bacterium. For instance, in the model host *Anaeramoeba*, some symbionts present within the host are considered as ectosymbionts as they are housed in a membrane network with connections to the plasma membrane (Jerlström-Hultqvist et al. 2024, Nat Commun, [10.1038/s41467-024-54102-7](https://doi.org/10.1038/s41467-024-54102-7)). In this study, no comparable observations were noted. Instead, only endosymbiotic bacteria completely residing within the cytoplasm were observed. To make the sentence clearer, we split this sentence in two, to read **line 262, page 7**: “All bacteria appeared to be completely enclosed within their host without connection to the plasma membrane and the extracellular media, they were then considered as endosymbionts. The bacteria were rod-shaped and, when observed outside the disaggregated protist, flagella were never observed.”

Line 323: should read ‘burnetii’

Response: The correction has been made.

Line 326: unclear what is meant with ‘the same ecological niche’, please clarify.

Response: We specified that the 16S rRNA gene sequences closely related to GD-1 were isolated from various aquatic habitats. We modified the sentence **line 331, page 8**, by: “Few close 16S rRNA gene sequences of uncultured bacteria associated to various freshwater and saline habitats, could be retrieved from public databases (**Supplementary Fig. 8d**).”

Line 336-337: It is not obvious to me that the Desulfovibrionales in the CLSM picture is at the anterior part of the cell, as mentioned in text. Rather it seems to stretch across the whole length of the host. As the FISH image in Fig. 6a and in Video 8 are the same cell, the authors should include a few more representative FISH images to support the claims about symbionts' location and abundance.

Response: Indeed, the *Desulfovibrionales* appear to stretch across the whole length of the host. We wrote in the text that these biomineralizing cell “were generally located at the anterior pole” which is not confirmed by the CLSM picture. This pattern of cell distribution was observed in several holobionts. However, to avoid the discrepancy with some of the images we decided to show in the manuscript, we modified the sentence **line 341, page 8**, by: “*Bacteria producing magnetosomes bundles were stretched across the whole length of the host as shown in Supplementary Videos 5, 8 and 9, and Fig. 3a.*”

In the new **Supplementary Video 9**, we now present three different holobionts that confirmed symbionts' location and abundance.

Line 343-345: From this one FISH picture it seems to me that the gammas are in fact more abundant than the Desulfovibrionales, in contrast to what is stated in text.

Response: We agree with this reviewer. Similarly to the positioning of the *Desulfovibrionales*, the abundance of the *Gammaproteobacteria* but also of the *Alphaproteobacteria* symbionts was not consistent between holobionts. This sentence, **line 353, page 8**, in the revised version, has been modified by: “*In contrast, the gammaproteobacterium GD-1 and alphaproteobacterium MD-1 lack specific spatial localization.*”

Line 351: what is meant with ‘symbiotic counterparts’, who is this referring to?

Response: ‘symbiotic counterparts’ was used for the *Desulfovibrionales*, *Alphaproteobacteria* and *Gammaproteobacteria* symbionts. We understand the confusion here. To make the sentence clearer it has been modified **line 357, page 9**, by: “*Despite its reduced size, the genome of DcD-1 has a relatively high G+C content (60.5%) compared to the other three genomes, whose G+C contents are significantly lower (from 37.2% to 45.7%, Supplementary Table 1).*”

Line 354: I find a protein-coding density of 57% very low and rather atypical for an endosymbiont – the authors could touch on this and the high pseudogene presence in their discussion.

Response: We agree, this is an interesting result that should be discussed. The following sentence has been added **line 597, page 13**, in the revised manuscript: “*The association of GD-1 and MD-1 with their ciliate host appears ancient as their genome structures reflect an advanced stage of reductive evolution. However, further investigation is required to date the relative emergence of this symbiosis in their respective taxonomic groups. Indeed, although families of the Rickettsiales order are considered to be relatively old groups of intracellular bacteria^{58,83,84}, associations with particular hosts may have arisen more recently. On the other hand, UBA9339 symbiont resides in a poorly explored group of orders related to Legionellales and Coxiellales. While there are suggestions that host association in general*

may be ancestral in these lineages, even its obligatory nature may be a more recent trait⁶³. Despite its relatively large genome size (>2 Mbp), the magnetic symbiont DsD-1 shows clear signatures of a long-term obligate symbiotic lifestyle as well. DsD-1 contains very few mobile genetic element–related genes and almost no pseudogenes, and it has a minimal set of conserved genes for DNA replication recombination and repair⁵⁴. However, in such advanced reductive evolutionary processes, one would not expect the unusually low protein-coding density observed in DsD-1 (57.43%), which indicates that nonfunctional sequences have not yet been fully purged. This genomic architecture suggests that DsD-1 may be undergoing an advanced, deletion-driven stage of genome reduction that followed the earlier loss of MGEs and pseudogenes. In contrast, the symbiosis establishment could have been more recent for the *Desulfarculales* endosymbiont whose genome is highly pseudogenized (6 %) and contains more than 1.5 k MBEs. Such a genomic structure could be the result of, or a remnant of ongoing reductive evolutionary processes following endosymbiosis establishment, as observed in other symbiotic bacteria^{54,74,84,85}.

Line 358: 'typical of obligate endosymbionts' is vague and unclear what this refers to. Be more concrete or remove such generic statement.

Response: We removed this generic statement.

Line 360: please include how many % of all proteins these three categories amounted to.

Response: We added this information **line 367, page 9**, in the revised manuscript.

Line 366: 'apart from Dsd-1, a last category outcompeted all the others:...' – what do you mean with this? Please revise and split these two statements into two sentences for better readability.

Response: We modified this sentence for better readability, **line 372, page 9**, by: "The COG category "Replication, recombination, and repair" represented 14%, 16%, and 30% of the predicted proteins in the endosymbionts GD-1, MD-1, and DcD-1, respectively. Of these, 46%, 63%, and 94% corresponded to mobile genetic element–related genes. In contrast, such genes were rare in DsD-1, where this COG category accounted for less than 0.02% of all annotated proteins."

Line 372: Who are the *Pseudomonadota* symbionts? In the diversity description *Pseudomonadota* are not mentioned.

Response: Here the *Pseudomonadota* symbionts are the *Alphaproteobacteria* MD-1 and the *Gammaproteobacteria* GD-1. Strains DcD-1 and DsD-1 belong to another phyla, the *Desulfobacterota*. This sentence has been totally modified based on a comment by reviewer #2. To avoid any ambiguity, we now specify the strain name whenever a taxon is mentioned throughout the revised manuscript.

Line 382: Please use the taxonomic names for each symbiont consistently throughout the text and do not switch between e.g. GD-1, Legionellales, Gammaproteobacteria when describing the same organism as this complicates the reading.

Response: The modification has been made throughout the Results and Discussion.

Line 406: do you mean oxidation of organic compounds (such as acetate)? Or really degradation (of polymers, such as PHB or such)?

Response: We agree that this statement may not be detailed enough. We rephrased **line 413, page 10**, as follows: "...the analysis indicates possible syntrophies based on the

degradation of organic compounds such as acetate or glucose, the oxidation of hydrogen and sulphate reduction...”

Line 413: do you mean it was not complete (rather than not fully predicted) in the genome?

Response: Indeed not all genes involved in the Wood-Ljungdahl pathway were predicted, we modified the sentence, now **line 422, page 10**, by: *“although this pathway was not complete in the draft genome.”*. We also included the KOFamScan outputs in a new **Supplementary Table 10** listing genes with KEGG Orthology annotations.

Line 441ff: I don’t understand this sentence. This study does not report the euglenozoan holobiont. Please revise.

Response: We start the discussion by a short reminding of the discovery in 2019 of the first collective magnetotaxis between an euglenozoan host and tens of ectosymbionts that biomineralize magnetite. We modified these sentences (now **line 453, page 10**, in the revised manuscript) to improve clarity at the beginning of the Discussion.

Line 446: I’d recommend to remove the ‘five-member’ as it is not clear if other symbionts other than the Dsd-1 are necessary for functioning of the consortium and also 5 member reference omits the diatom.

Response: The modification has been made.

Line 458: ‘Unfortunately, ...’ What is meant with this sentence? It does not make sense here, I suggest removing.

Response: We aimed to mention that we hoped to get a more resolved tree of the Ciliophora with more markers, but that we could not because no eukaryotic DNA was amplified or host genome assembled. To clarify, we rephrased as follows **line 470 page 11**: *“The 18S rRNA gene sequence might not be resolute enough to get a proper insight into the relationships with other ciliates. The tree topology lacks support and differs with those reconstructed from numerous orthologs⁷². A more resolved phylogenetic reconstruction was not possible as no genome associated with the host could be assembled from the amplified DNA of the holobiont.”*

Line 484: Please include the number.

Response: This sentence has been modified, **line 500, page 11**, by: *“This number of magnetosome chains far exceeds the minimum number of few dozen magnetosomes required to generate sufficient torque in such MHB to align with Earth’s magnetic field lines³².”*

Line 496: ‘should be largely enough’. As you document magnetotactic behavior of the ciliate, it is clearly enough. Please revise accordingly.

Response: This sentence, now **line 503, page 11**, has been modified by: *“In total, approximately 800 magnetosomes are produced by E. magneticus DsD-1 cells from a single host to induce a magnetotactic behaviour in the ciliate.”*

Line 503: reminding how? Please amend the statement to be more concrete.

Response: We agree on this comment but this sentence has been removed, as Reviewer #2 considered it too speculative.

Line 528: I am not sure I follow why the discovery in France and Brazil would raise question about the presence of magnetite biomineralization in microeukaryotes. Do you maybe mean biogeography or distribution or origin? Please be more specific.

Response: To date, the only microeukaryote proposed to biomineralize magnetite is the magnetotactic protist described by Leao et al. (2019). However, our study shows that this protist is actually in symbiosis with magnetite-biomineralizing bacteria, indicating that no microeukaryote is currently known to biomineralize magnetite. Nonetheless, it remains possible that undiscovered microeukaryotes with this capability exist. We rephrased this sentence, now **line 541, page 12**, by: "*Magnetite biomineralization by microeukaryotes still need to be confirmed.*"

Line 530: Revise this formulation – I guess you mean association of protist host from different lineages with symbionts from different phyla?

Response: We agree, it is clearer to formulate the sentence that way. It has been modified, **line 542, page 12**, in the revised manuscript, as follows: "*The association of protist hosts from diverse phyla with symbionts from various classes suggests that magnetosymbiosis has independently emerged multiple times across both the Bacteria and Eukaryota domains.*"

Line 532-533: It should also be considered here that the emergence of these symbioses will be likely facilitated in environments where magnetotactic prokaryotes are prevalent and common.

Response: We agree, this notion is relevant in this paragraph, now **line 544, page 12**. We added the following sentence: "*Moreover, the emergence of these symbioses will be likely facilitated in environments where magnetotactic prokaryotes are prevalent and common.*"

Line 539: should be 'consuming' not 'producing'

Response: Modification has been made, now **line 557, page 12**.

Line 542 - 547: which other hydrogen-producing specialized organelles are meant here? The protist can utilize 'external inorganic electron acceptors'? Which would these be? 'Offload' sounds a bit like slang. Also, hydrogen doesn't need to be 'offloaded', it will diffuse out of a swimming ciliate without problems. What about the sulfide that is the end product of the Desulfobacterota metabolism – how does the host deal with that toxic substance being produced intracelullarly?

Response: Concerning the "other hydrogen-producing specialized organelles" it was not meant to point out a specific organelle but rather suggest the possibility that other organelles could produce hydrogen. We simply removed these words.

We agree that this whole section was not really well written. We rephrased **line 559, page 12**, in the revised manuscript as follows: "*Predatory protists metabolize organic matter and produce hydrogen (H₂) as a byproduct via their hydrogenosomes and small molecules such as CO₂, acetate or alcohols. In the magnetotactic ciliate, these intermediates are likely further oxidized by their endosymbionts for biomass and ATP generation. On the other hand, DcD-1 and DsD-1 could enable their host to breathe sulphate instead of oxygen. Such utilization of symbionts as electron acceptors or sinks in metabolic coupling has been*"

previously documented in other ciliates⁸¹. In addition, hydrogen sulphide produced by sulphate respiration of DcD-1 and DsD-1 could help maintain a low redox potential inside the consortium or could be eliminated outside the protist by diffusion to avoid toxic effects.”

Line 567: I would not say that Rickettsiales are better known for mutualism as many people (including me) might associate them with human pathogens. Or do you mean the specific group that includes your symbiont?

Response: Indeed, in this sentence we are discussing about the specific family to which belongs MD-1. In this sentence, now **line 588, page 13**, we specified that we are talking about the *Midichloriaceae* family.

Line 570: What is MHB? I missed this explanation.

Response: MHB referred to “magnetotactic holobionts and was defined **line 283, page 7** (in the revised manuscript), when the acronym appears for the first time. This acronym is used numerous times throughout the text. This acronym was also previously used throughout the manuscript Chevrier et al 2023, PNAS.

Line 571: Spell out whose genome.

Response: This paragraph **line 589, page 13**, was modified for clarity.

Line 575: Which associations and lost from where, which genomes/organisms?

Response: We agree, it was unclear if we were talking of our model or of the example from the previous sentence. In fact, we are discussing about our model, we specified this point in this sentence to avoid the ambiguity. **Line 597, page 13:** “*The association of GD-1 and MD-1 with their ciliate host appears ancient as their genome structures reflect an advanced stage of reductive evolution.*”

Line 582: I would not describe being prey as a facultative relationship – or what is meant here?

Response: This sentence, now **line 621, page 14**, was modified and the term “facultative relationship” was removed: “*The presence of different diatom species within the same population of magnetotactic ciliates from a single sample supports the idea that intracellular diatoms are not essential for the functioning of the ciliate or the holobiont.*”

Line 590-591: Is this the case in this host? Could you include a reference to a Figure then?

Response: Yes it does. It was observed in some of our thin sections such as the **Supplementary Figure 5e**. Based on a comment of Reviewer #2, this paragraph has been shortened (from line 584 to line 619 in the former manuscript) and this sentence has been removed.

Line 598 – 599: Also, I assume that sediments will not be limited in any of these nutrients, unlike the water column.

Response: We agree that sediments should be less limited than the water column in CO₂, inorganic nutrients, such as phosphorus, nitrogen, or trace elements. However, we do not know if the diatoms found in the ciliate were living in the water column or the sediments before their engulfment. Based on a comment of Reviewer #2, this paragraph has been shortened and this sentence has been removed.

Line 599ff: I wonder - if the diatom (or its chloroplasts) are used by the host for photosynthesis what happens with the produced oxygen? Can you speculate how the obligately anaerobic sulfate reducers can possibly deal with this?

Response: The relationship between the diatoms and the ciliate is only hypothetical here and we hope that our future research will allow to decipher what exactly linked both microeukaryotes. Here the ciliate appears to be living in anaerobic conditions but it might also be able to respire oxygen as most of ciliates do. Thus, it is possible that the oxygen produced by the diatom during photosynthesis is directly used by the ciliate and that prokaryotic endosymbionts are not in contact with this oxygen. It is also possible that the oxygen diffuse outside the ciliate and its time inside the ciliate is limited to avoid potential toxic effect. In this case the ciliate would benefit only from the energy produced during photosynthesis. We could also hypothesize that the host is not the only one to benefit from the oxygen released by the diatom, the bacterial symbionts could also use oxygen as a terminal electron acceptor, even the *Desulfobacterota* as some strains of the *Desulfovibrionia* have been shown to be able to respire oxygen, including *Solidesulfovibrio magneticus* (doi: 10.1111/1758-2229.12479), a magnetotactic bacterium related to strain DsD-1. However, based on current assemblies, it is unlikely as no cytochrome *c* or *cbb₃* oxidases were identified (**Supplementary Table 9**).

We wanted to add the sentence: "The oxygen produced during photosynthesis could also benefit to the bacterial symbionts even the *Desulfobacterota* as some have been shown to grow with oxygen⁸³. Alternatively, the oxygen could be toxic for the host and the symbionts once released inside the ciliate, its rapid diffusion outside the ciliate could potentially avoid damages." However, the whole paragraph has been removed as the Reviewer #2 found it too speculative at this stage.

Line 652: Care should be taken to clarify that in the euglenozoa symbiosis magnetotaxis of the ectosymbiont is combined with the chemotaxis (aerotaxis) of the host, and thus allow navigation in chemically-stratified aquatic environments. Is this ciliate host also capable of aero- or chemo-taxis?

Response: We do not have evidence of host ability of aero- or chemo-taxis in both models of magnetotactic protists. It is only an hypothesis to explain the potential advantage conferred by magnetotaxis based on what has been demonstrated in free-living magnetotactic bacteria. The sentence was modified to the conditional, **line 656, page 14**: "*The magnetotaxis observed in the magnetotactic holobiont HBD-1 from the Dordogne river could mirror that of free-living magnetotactic bacteria and euglenozoan, enabling efficient navigation in chemically stratified aquatic environments*"

Line 657: This is to me an unclear statement and it is a pity to end up this really nice paper on this kind of vague sentence. Maybe try to revise to be more specific.

Response: In this sentence we meant that the evolution of collective magnetotaxis based on an endosymbiosis between a ciliate and biomineralizing bacteria could be at the origin of magnetoreception in eukaryotes. We agree that we should have been more specific in this sentence. We modified this sentence, **line 661, page 14**, by: "*However, although the integration of magnetotactic bacteria into a modern lineage of protists is now proven, further research is needed to determine whether such integration and further organellogenesis or lateral gene transfers could have occurred in early holobionts that gave rise to the first eukaryotes lineages able of magnetoreception.*"

Paragraph Line 675ff: What is meant by microcosms? Please explain. It is unclear to me how

the oxygen sensor was calibrated – what is ‘saturated humid air (O₂ saturation 100%)’? Do you mean pure oxygen? What is humid? Why was the calibration not done with air-saturated water, as outlined in the instruction manual? What is ‘water solution flushed with N₂’? Is it MilliQ? Was it flushed or bubbled? How long? Please include all relevant information here.

Response: Microcosms is commonly used to mean mesocosms or aquaria. Since “mesocosms” seems to be more often used, we replaced “microcosms” by “mesocosms”.

The calibration of the oxygen probe was made with 1) a 50 mL solution of milliQ water flushed (i.e. bubbled) for 30 min with air in order to get 100% of oxygen saturation = 21% oxygen in the water. “humid” is generally employed to show that the calibration is made in a liquid, here milliQ water; 2) the second point of calibration was made in 50 mL milliQ water flushed (i.e. bubbled) for 30 min with 100% nitrogen. This information have now been included in the Methods section, **line 687, page 15**, in the revised manuscript: “*Sensor calibration was made with a 50 mL milliQ water solution flushed with air for 30 min (O₂ saturation = 100%) and a milliQ water solution flushed with N₂ for 30 min (O₂ saturation = 0%).*”

Paragraph Line 788ff: It is essential to include here the formamide concentrations of the hybridization buffers, which were used for FISH.

Response: The stringency used during FISH protocol was already indicated in the Methods: “*using the hybridization and washing stringencies of 30% for all probes*”. In general, we do not need to indicate the formamide concentration since it is proportional to the stringency. Here a stringency of 30% mean that the formamide concentration in the hybridization buffer was 0.6 mL/mL. We added this information in the revised Materials and Methods section **line 807, page 17**.

Line 1260: replace ‘blows’ with ‘breaks apart’

Response: Modification has been made **line 1310, page 30**, and also in the caption of **Supplementary Video 3**.

Line 1279: replace ‘presence of the extremity of a diatom’ with ‘part of a diatom’.

Response: Modification has been made **line 1329, page 30**.

Line 1325: replace ‘authentication’ with ‘identification’ or ‘visualization’

Response: Modification has been made. We preferred to use “visualization” **line 1375, page 31**.

Line 1328: replace ‘authenticate’ with ‘target’

Response: Modification has been made **line 1378, page 31**.

Line 1337ff: Please revise this sentence for clarity. Also, sulfate reducers typically do not use more complex organic compounds.

Response: The sentence, **line 1388, page 31**, has been modified by: “*Complex carbohydrates such as glucose would benefit to the endosymbiotic Gammaproteobacteria and Alphaproteobacteria that could feed from the primary breakdown of organic material in digestive vacuoles.*”

Line 1340-1341: replace with ‘... and might depend on the biosynthetic capacities of other symbionts for their growth’.

Response: The legend of **Figure 6b** has been modified, this sentence does not exist anymore, although the information is still present *line 1393, page 31*.

Reviewer #2 (Remarks to the Author)

The paper entitled "Magnetoreception in a freshwater anaerobic ciliate arises from endosymbiosis" represents the first documented case of magnetoreception in a protist resulting from an endosymbiotic association with bacteria. While magnetoreception in protists remains a rare and poorly understood phenomenon - previously observed only in protists with ectosymbionts, this study reveals an astonishingly complex and novel system. The discovery not only expands our understanding of magnetoreception in eukaryotes, but also contributes to the growing body of research highlighting the remarkable diversity and evolutionary significance of interactions between bacteria and protists.

As the first discovery of a system in which magnetoreception is acquired through endosymbiosis, this research is both novel and timely and fits well into the current wave of studies investigating endosymbiotic interactions between bacteria and protists. The results are thoroughly documented and supported by multiple lines of evidence, including advanced microscopy techniques and comprehensive genome analyses.

Response: We thank the reviewer for their kind comments. Their comments not only strengthened the manuscript but also broadened our personal understanding of protist symbioses.

However, there are concerns regarding the interpretation of the results. While the involvement of the endosymbiont DsD-1, a member of the Desulfovibrionaceae, in magnetoreception is well supported, the specific contributions of other members of the holobiont, both bacterial and eukaryotic, remain unclear. These additional partners may play an important role in supporting the anaerobic lifestyle and overall fitness of the host, but their direct involvement in magnetoreception has not been convincingly demonstrated. Therefore, characterising the phenomenon as “magnetotaxis of a multi-partner endosymbiotic interaction” appears premature based on current data and their interpretation.

The manuscript repeatedly emphasises that magnetoreception arises from the interaction between several partners; however, the evidence presented does not sufficiently support this claim. Either more direct evidence is needed to demonstrate the involvement of all partners in magnetoreception, or the current formulation should be weakened to more accurately reflect the limitations of the data. Ideally, experiments on cultures would clarify the critical role of individual partners, although this is often difficult to achieve. Therefore, a more detailed reconstruction of metabolism that emphasises the interdependence of all holobiont components is required, together with more careful interpretation.

Response: We agree with the reviewer #2: we believe that magnetoreception in the protist is possible only thanks to one of the four endosymbionts, DsD-1. We apologize for the awkward expression “magnetotaxis arises from a multi-partner endosymbiotic interaction” that led to this confusion in the abstract and suggests that several symbionts are involved in magnetoreception. In the revised manuscript, we comprehensively specified the organisms involved in magnetotaxis when describing the holobiont.

For example, we reworded the expression in the abstract **line 42, page 2**, by simply removing the term “multi-partner” as follows: “*However, in this case, magnetotaxis arises from an endosymbiotic interaction.*” The origin of magnetoreception was also specified in the introductory section **line 139, page 4**: “*We demonstrated that these unicellular eukaryotes are, in fact, multi-partners holobionts composed of a ciliate (Ciliophorea, Prostomatea) and four populations of endosymbiotic bacteria, one of which, belonging to the Desulfovibrionales, biomineralizes a bundle of bullet-shaped magnetosomes.*”

Detailed comments on these and other issues are provided below. Overall, the study presents a convincing result that is supported by solid data. However, our main concerns relate to the way in which the results are communicated. In some cases, the claims are presented too broadly, and in others, the interpretations appear too speculative. We believe that a more focused presentation on the magnetoreception aspect would strengthen the manuscript.

Response: Based on the corrections proposed below by this reviewer, we believe that the presentation of the results and the discussion is now more focused on the magnetoreception aspect. In addition, the entire discussion section from line 584 to line 619 on the relationships of diatoms with the ciliate was removed as it was found too speculative.

Detailed comments

Line 42 – “However, in this case, magnetotaxis arises from a multi-partner endosymbiotic interaction.” Is the magnetoreception in the case presented strictly dependent on multiple endosymbionts (instead of just single magnetotactic symbiont), in contrast to previously known magnetoreception in euglenozoans (relying on extracellular bacteria)?

Response: Indeed, this sentence is confusing. As mentioned above, we removed the term “multi-partner” **line 42, page 2**, in the revised version.

Line 65 – There are known cases of endosymbionts residing in the nucleus (which formally is distinguished from the cytoplasm) (10.1016/j.tcb.2015.01.002), thus it is better to change for the “integrated within the host protoplasm” or “integrated within the host cytoplasm and/or nucleus”.

Response: A similar comment was raised by Reviewer #1, we replaced “integrated within the host cytoplasm” by “integrated within the host cell” **line 64, page 3**, in the revised version.

Line 104 - please change protozoa to protist

Response: Modification has been made **line 103, page 4**, in the revised version.

Line 105 - please use mitochondria-related organelles (MRO) as a well-established term for reduced mitochondria in anaerobic lineages of protists

Response: Modification has been made **line 104, page 4**, in the revised version.

Line 110 – rather “protection against competitors or predators”, not “chemical camouflage” (the latter not mentioned in the original publication).

Response: Modification has been made **line 109, page 4**, in the revised version.

Line 204 - In eukaryotes, genetic distance based on 18S rDNA is generally not sufficient to make reliable taxonomic decisions, as evolutionary rates vary greatly between lineages. The guidelines of the Zoological Code should be followed when describing new ciliate species. The data presented may not be sufficient to formally describe a new species or genus, as this requires the identification of type material. Furthermore, a formal taxonomic description does not appear to be necessary in this case, and its absence would not diminish the significance of the results. We therefore suggest reconsidering whether such a description is needed.

Response: We agree that further work would be required to give a formal taxonomic description to this protist. We replaced the statement *line 214, page 6*, by “*Pending a more formal taxonomic description in the future, we named the ciliate HBD-1*” in the revised version. In addition, we removed the reference to *Candidatus Magnetocilia dordonia* throughout the manuscript when referring to the host and replaced the proposed genus and species name by the term “magnetotactic protist”, “ciliate” or “HBD-1”.

Line 287 - we suggest providing the information on the average genome coverage for the obtained bacterial genomes. This will better illustrate the quality of data presented and may inform about relative abundances of the symbionts in the host cell.

Response: We added this information as requested in the new **Supplementary Table 1**.

Line 288 - how the estimation of the amount of eukaryotic reads was performed? As the authors attempted total rRNA extraction from the raw assembly, is it correct to state that no 18S fragments of either host or intracellular photosynthetic algae were recovered? If so, please mention this in an appropriate section of the results.

Response: The number of eukaryotic reads was estimated as follows: first, EukRep was used to identify eukaryotic contigs within the assembly. Then all reads were mapped to these contigs with Bowtie 2 to estimate the number of eukaryotic reads. This information has been added in the materials and methods section *line 857, page 18*, in the revised manuscript.

The rRNA genes were predicted using RNAmmer in all the global assembly before binning as stated *line 871, page 18*, in the revised manuscript and led to the identification of only one 18S rRNA gene sequence matching with the 18S rRNA sequence of the ciliate obtained by pyrosequencing. No other sequence was recovered.

The corresponding result section was rewritten and corrected as follows: “*However, only few eukaryotic reads were obtained after the hologenome amplification— approximately 31 million, representing 15.3 % of the total reads. They mapped to 2 209 contigs representing a total assembly length of 19.95 Mbp. According to the Protista_83 single-copy core gene (SCG) collection in anvi'o, the assembly corresponds to a highly incomplete genome (< 6.0% completeness) with substantial redundancy (> 9.6%). Only a single 18S rRNA gene sequence associated with the ciliate was detected.*” *line 289, page 7*, in the revised manuscript.

Lines 298-299 - as Dsd-1 is currently the only endosymbiotic magnetotactic bacterium in its family, authors may consider comparing its genomic capabilities with that of the free-living relatives (*Oceanidesulfobrio* and *Megalodesulfobrio*). Similarly, comparison with *Ca. Desulfarcum epimagneticum* CR-1 may inform about the differences in the integration between eukaryotic hosts and their magnetotactic symbionts.

Response: We fully agree with Reviewer #2. We initially began conducting this broader analysis for this symbiont as well as others, including some unpublished ones. However, we soon realized that the volume of data and the number of insights were substantial enough

to write a dedicated manuscript of their own on the emergence of a symbiotic lifestyle in magnetotactic *Desulfobacterota*. For the present manuscript, we believe it is more appropriate to focus specifically on magnetoreception.

Lines 339-341 - Presence of the mam genes is mentioned in support of the ability of Dsd-1 to form magnetite particles. While this proof seems sufficient, Extended data Fig. 8e shows that some of the core mam genes (according to Lefevre et al. 2013 <https://doi.org/10.1111/1462-2920.12128>) are missing (mamA, mamE, mamQ, mamB). Was there an attempt to find those genes in the assembly in order to confirm or rule out that their absence was caused by binning inaccuracy?

Response: This is correct. As stated in the original manuscript, the MGC is incomplete. The absence of the mentioned *mam* genes is likely due to amplification, sequencing, or assembly issues, as the MGC lies at the end of a truncated contig. These genes were not detected elsewhere in the genome. The remaining 27 genes are conserved in deep-branching MTB and appear to be specific to magnetotaxis. Additional sequencing projects are planned to improve the assembly of the MGC.

Additionally, while unnecessary here, the phylogenetic analysis of mam genes would be instrumental in understanding the process of biomineralisation cluster acquisition.

Good point. A phylogenetic analysis of *mam* genes or Mam proteins would be very interesting indeed, and is complementing the comparative genomics analysis mentioned above. As the reviewer noted, several *mam* genes that are among the most conserved in MTB (see doi:10.1038/nrmicro.2016.99) are missing, which currently limits the robustness of phylogenetic inference. However, we agree that we could have provided more information regarding the potential origins of these genes.

Based on BLASTP alignments, Mam proteins of DsD-1 share the highest sequence identity (i.e., 50.6 % and 47.7 % on average) with their corresponding homologs in the reference strains *Solidesulfovibrio magneticus* RS-1 (<https://doi-org/10.1101/gr.088906.108>), and *Fundidesulfovibrio magnetotacticus* FSS-1 (<https://doi-org/10.1099/ijsem.0.005516>), respectively. Together DsD-1, RS-1 and FSS-1 represent the only three genera containing magnetotactic species (**Figure 5a** and the new **Supplementary Table 2** in the revised manuscript). A phylogenetic tree of the most conserved Mam protein among the all-shared Mam proteins, MamK, supports that the MGC of RS-1 and FSS-1 have a more recent common ancestor than both RS-1 and FSS-1 MGC with that of DsD-1. Such topology can support several scenarios given the species tree in **Figure 5A**, involving and not, ancient horizontal gene transfers—even if the first scenario (with HGT) is the more parsimonious. In the revised manuscript, we simply specified the relatedness of Mam proteins with their closest MTB, **line 348, page 8**: “Mam proteins of DsD-1 share the highest sequence identity with their corresponding homologs in the reference strains *Solidesulfovibrio magneticus* RS-1⁵², and *Fundidesulfovibrio magnetotacticus* FSS-1⁵³, with average identities of 50.6 % and 47.7 %, respectively.”. In addition a new supplementary table was provided showing these identities as proposed by the reviewer #3 (**Supplementary Table 2**)”.

Line 354 - How were the pseudogenes identified? This should be included in the methods.

Response: Pseudogenes were identified using the Microscope platform (<https://doi.org/10.1093/database/bap021>), based primarily on ortholog length comparison. Each CDS of the symbionts was aligned to its ortholog in *Escherichia coli* K-12 substr. MG1655 (GCF_000005845.2). Cases in which a single *E. coli* CDS corresponded to two colinear ORFs in the symbiont genomes were interpreted as gene fragmentation, typically

resulting from frameshift mutations or premature stop codons. Because no closely related complete genomes were available for comparison, the extent of pseudogenization reported here is likely conservative. This clarification has been added to the Methods section (**line 864, page 18**, in the revised manuscript): “*Pseudogenes were detected by aligning each symbiont CDS to its ortholog in E. coli K-12 substr. MG1655 and identifying orthologs with disrupted structure. A gene was classified as fragmented when a single E. coli CDS matched two colinear ORFs in the symbiont genomes, indicative of frameshift mutations or premature stop codons.*”.

Lines 364-366 – we find the information about the proportion of proteins related to glycosyltransferases a bit out of context, as it is not interpreted or mentioned further. Is this proportion different in other symbionts or somehow remarkable in this case?

Response: We are sorry, we can't justify this statement in the original version. We agree with the reviewer, and we removed the sentence in the revised manuscript.

Lines 366-368 - there is an enormous number of replication, recombination & repair genes identified in genome Dcd-1 (as shown in Supplementary Table 1), an order of magnitude larger than in other symbionts or free-living bacteria used for comparison. Is it possible to assess whether this inflated number is not caused by the mobile genetic elements-related genes? They often fall within the same COG category and may contribute to the assembly fragmentation. In addition, their identification informs about the proliferation of mobile genetic elements, observed in some symbiont genomes.

Response: Excellent point, we missed it in the original manuscript. Indeed, the number of replication, recombination, and repair genes is highly inflated in DcD-1 due to mobile genetic element-related genes, which we had initially overlooked. To quantify their contribution, we screened the eggNOG-mapper output for COG category “L” and identified genes whose PFAM domains or annotation descriptions contained the terms “IS,” “transposase,” or “integrase” (see the new **Supplementary Table 4**). Mobile genetic elements accounted for 94%, 63%, 46%, and 4% of category L predicted genes in DcD-1, MD-1, GD-1, and DsD-1, respectively. This observation and interpretation have been added to the revised manuscript, **line 374, page 9**.

Lines 368-371 - the loss of the replication, recombination & repair machinery in Dsd-1 is very interesting, as those processes are highly conserved and mostly retained even in extremely reduced endosymbionts with genomes below 1 Mbp (10.1146/annurev-micro-091213-112901 and 10.1371/journal.pbio.3002577). Therefore, we suggest presenting more information on the components that are preserved or lost, and whether this loss would impact the bacterial capability to maintain its genome. It is important information for understanding the genome reduction processes in endosymbionts.

Response: We apologize, this statement was not correctly written, because indeed, it suggests that ALL conserved genes were lost and this is not correct. We meant that DsD-1 machinery was reduced. We carefully checked the eggNOG-mapper output for COG category “L” (see the new **Supplementary Table 4**), and we confirm that a minimal replication, recombination & repair machinery is present in all symbionts. We corrected this sentence and provided more information on the genes preserved in the results section **line 376, page 9**, in the revised manuscript: “*All symbionts have a minimal machinery involved in DNA replication, homologous recombination, base excision repair or mismatch repair. The set of conserved genes includes polymerases genes together with dna, rec, ruv, uvr, and mut genes (Fig. 5e and Supplementary Table 4)*”.

And we discussed this observation and its meaning **line 605, page 13**, in the Discussion section of the revised manuscript: “Despite its relatively large genome size (>2 Mbp), the magnetic symbiont DsD-1 shows clear signatures of a long-term obligate symbiotic lifestyle. DsD-1 contains very few mobile genetic element–related genes and almost no pseudogenes, and it has a minimal set of conserved genes for DNA replication recombination and repair⁵⁴.”.

Line 371 - We suggest adding a reference to the Supplementary table 1 as well, because Fig. 5e shows the relative frequencies of gene categories and not the absolute number of genes, which is important for interpreting the gene losses of conserved cellular systems.

Response: Reference to this Supplementary table has been added, in the new version of the manuscript it is **Supplementary Table 4**.

Line 372 - Proteins with typical eukaryotic-interacting domains (including leucine-rich repeats) are presumed to participate in symbiont manipulation of the host cell after delivery via secretion systems (10.1016/j.cub.2021.05.049). While it is possible that they were acquired via HGT, there is no direct evidence for that presented, and we would suggest focusing on their potential role in interaction with the host. In particular, we suggest providing information regarding the number of such genes for a particular symbiont genome.

Response: We agree. First, we removed the sentence “Genomes of both *Pseudomonadota* endosymbionts specifically, show evidence for lateral gene transfers.”. Second, we built a list of proteins containing leucine-rich repeats (LRRs) and ankyrin repeats—domains involved in interactions with eukaryotic hosts (new **Supplementary Table 5**). Homology searches were performed using the The InterPro database (<http://www.ebi.ac.uk/interpro/>) integrating together predictive models or ‘signatures’ representing protein domains, families and functional sites from multiple, diverse source databases: Gene3D, PANTHER, Pfam, PIRSF, PRINTS, ProDom, PROSITE, SMART, SUPERFAMILY and TIGRFAMs. Finally, we reported the number of such genes in symbiont genomes **line 379, page 9**: “Functional annotation predicted genes encoding proteins with leucine-rich repeats (LRRs) and ankyrin repeats—domains involved in interactions with eukaryotic hosts⁵⁸—in three out of four symbiont genomes (**Supplementary Table 5**). While the DcD-1, GD-1, and MD-1 genomes encode 3, 8, and 7 ankyrin repeat-containing proteins, respectively, only GD-1 harbors four proteins with leucine-rich repeats, alongside several Type IV secretion systems (**Supplementary Table 6**), which are often repurposed by endosymbionts to deliver host-targeted effectors⁵⁸”.

Line 387 - We suggest moving the details about the virulence factor search into the methods section.

Response: Modification has been made. In the Methods section, **line 926, page 20**, we added the following sentence: “Putative virulence-associated genes were identified using *VirulenceFinder*¹²⁵ and *BLASTP* searches against hits of the *Virulence Factor Database* (VFDB)¹²⁶ applying a minimal identity of 30%.”

Line 395-397 - We suggest toning down this interpretation of function as many of the presented virulence factor predictions are based on relatively low (30%) identity hits and may indicate only distant similarity and not the same function.

Response: We fully agree. The following section was rephrased **line 398, page 9**, as follows: “A total of 118 genes in GD-1 showing remote homology to known virulence factors were also predicted, including a *phtA*-like gene, a phagosomal transporter involved in the establishment and persistence of *L. pneumophila* within its replication vacuole⁶⁵, as well as

enhanced entry (Enh) proteins that facilitate host cell invasion and infection⁶⁶. Many transporters involved in iron uptake, adherence and toxins (e.g. Enterobactin, Hemolysin, LPS, Microcin H47) in pathogens such as *Neisseria meningitidis* MC58, *Escherichia coli* O157:H7, *Haemophilus influenzae* Rd KW20, were also predicted. However, some sequence identities being low (*i.e.*, 30-40%), these homologs may be functionally different. No such genes were detected in MD-1, except antophagocytosis genes, like the two other symbionts exhibiting similar but less numerous defensive systems (**Supplementary Table 8**).”

Line 439 - We suggest rephrasing this, as there is no evidence for reliance of magnetoreception specifically on syntrophy, to the best of our knowledge. E.g. there is a possibility that those layers of interaction are uncoupled, or that magnetoreception was a driving force for the evolution of this association, with established syntrophy being a later byproduct.

Response: We clarified and simplified the sentence **line 455, page 10**, in the revised manuscript: “*This finding introduced the concept of collective magnetotaxis characterized by a mutualistic and obligatory symbiosis. The labour in magnetotactic holobionts is divided between partners: the euglenozoan host performs chemo-aerotaxis, while the ectosymbiotic bacteria provide magnetoreception.*”

Lines 474-476 - we strongly encourage authors to provide full, proper descriptions of the newly described bacteria (see examples, 10.1038/s41396-023-01499-6 and 10.1038/s41467-024-54047-x). Authors may also consider using SeqCode (10.1038/s41564-022-01214-9) for new taxa submission.

Response: We are convinced that providing proper descriptions of the newly identified taxa is highly valuable. The protologue has now been included in the **Supplementary Results 1** section. All names have been deposited in the SeqCode Registry under accession number r:dgcae3s5. Because the genome quality of DcD-1 was not sufficient, the curators recommended removing the name *Desulfella intracellularis* from the list and assigning it *Candidatus* status instead. The name will be formally published in SeqCode once higher-quality assemblies become available.

Lines 501-504 - presented statement suggests that the magnetite-forming endosymbiont has lost many of its metabolic capabilities, preserving only essential pathways for cell functioning. However, it still possesses a relatively large genome (2 Mbp) and in terms of the number of metabolic pathways identified is only the second lowest after the Midichloriaceae endosymbiont. Additionally, it is suggested to participate in syntrophy. We suggest to tone this down a bit or provide more details about the relevant pathway losses. For example, severe deficiencies of systems important for cell maintenance (phospholipid/fatty acids synthesis, peptidoglycan/cell wall synthesis, division, nucleotide and amino acid synthesis) may support this claim.

Response: We agree that this whole section was confusing. We deleted it in the revised manuscript. Instead, we focused on the magnetosome aspect **line 502, page 11**: “*Conversely, HBD-1 is larger than the marine MHB but harbours 25 times fewer endosymbiotic E. magneticus cells a total of approximately 800 magnetosomes to induce a magnetotactic behaviour in the holobiont. Despite their lower numbers, each cell produces almost five times more magnetosomes aligned into six chains of bullet-shaped magnetite crystals. These particles are nearly twice as large as those of free-living magnetotactic bacteria from the Desulfovibrionaceae family, such as Solidesulfovibrio magneticus strain RS-1⁷⁸ or Fundidesulfovibrio magnetotacticus strain FSS-1⁵⁴, which typically form about ten particles of similar width but less than 100 nm in length. However, they are smaller than*

those in a magnetotactic eukaryote described by Leão et al.³⁵, which had magnetosomes-like particles averaging 276.6 ± 61.3 nm in length and 52.7 ± 5.1 nm in width. Another striking difference between both magnetosome-producing symbionts is that the endosymbiont apparently evolved to overproduce magnetosomes to the point that they fill the entire cytoplasm. Indeed, cells of DsD-1 produce about 131 particles for a length and width of approximately $5 \mu\text{m}$ and $0.5 \mu\text{m}$, whereas cells of the cultivated strain FSS-1 produce about 10 magnetosomes for a length and width of approximately $4 \mu\text{m}$ and $0.8 \mu\text{m}$, respectively⁵⁴.”.

Line 569-570 - did authors consider performing a search for the ATP/ADP translocases in endosymbiont genomes? This information may be useful for assessing the potential impact of Pseudomonadota symbionts on the energy metabolism of the holobiont.

Response: We agree that the presence of ATP/ADP translocases in endosymbiont genomes is important to understand the nature of symbiont/host interactions. Homology searches were performed using the InterPro model IPR004667 and only three significant matches were identified in the alphaprotobacterium genome. This information is now included in a revised version of the **Figure 6b** and **line 434, page 10**, in the revised manuscript: “MD-1 has a severely reduced energy metabolism and lacks the ability to perform glycolysis and to use the tricarboxylic acid (TCA) cycle—similar to many Rickettsiales, which import metabolites from their host⁷⁰. However, MD-1 has the potential to generate reducing power (NADH), acetyl-CoA and CO₂ through the decarboxylation of exogenous pyruvate. It may also produce the (2S)-ethylmalonyl-CoA intermediate required for the ethylmalonyl-CoA pathway, an alternative to the TCA cycle. Notably, this endosymbiont is the only one in HBD-1 to encode three ADP/ATP translocases (*tlc*; InterPro entry IPR004667), which enable the direct import of ATP from the host cytosol⁵⁸.”. We also include this result in the Discussion section **line 591, page 13**, in the revised version.

Line 571-573 - we find this part slightly confusing, are those auxotrophies present in Pseudomonadota symbionts? If yes, how do they contribute to the consortium? Are there any contributions from the Desulfobacterota symbionts? Figure 6b uses the generalised term “amino acids” and only provides information about vitamins B1 (provided by all symbionts), B9 (provided by three symbionts) and B12 (provided by two symbionts), impairing interpretation.

Response: We agree that this part is confusing as we don't mention which symbiont has the inability to synthesize amino acid, vitamin, or nucleotides. We rephrased this section **line 591, page 13**, in the revised version as follows : «They likely do not participate directly to the global energy metabolism of the MHB and they are unable to synthesize a large number of compounds required for their growth. However, they may support the consortium by synthesizing essential compounds for both the host and other symbiotic partners, such as few vitamins and key cofactors». The **Figure 6b** was also revised and now includes the detailed list of amino acids/vitamins & co-factors synthesized by each bacterial partner.

Line 575-576 - Families of Rickettsiales are considered to be relatively old groups of intracellular bacteria, but associations with particular hosts may be more recent (10.1038/s41467-021-23645-4, 10.1038/s41467-024-45351-7 and 10.1016/j.cub.2021.05.049). On the other hand, UBA9339 symbiont resides in a poorly explored group of orders related to Legionellales and Coxiellales. While there are suggestions that host association in general may be ancestral in these lineages, even its obligatory nature may be a more recent trait (10.1038/s41564-022-01174-0). We suggest explaining this notion in more detail or citing a specific reference.

Response: We agree. We rephrased this section and included the following statement **line 597, page 13**: “The association of GD-1 and MD-1 with their ciliate host appears ancient as their genome structures reflect an advanced stage of reductive evolution. However, further

investigation is required to date the relative emergence of this symbiosis in their respective taxonomic groups. Indeed, although families of the Rickettsiales order are considered to be relatively old groups of intracellular bacteria^{58,83,84}, associations with particular hosts may have arisen more recently. On the other hand, UBA9339 symbiont resides in a poorly explored group of orders related to Legionellales and Coxiellales. While there are suggestions that host association in general may be ancestral in these lineages, even its obligatory nature may be a more recent trait⁶³.

Line 577 - The role of diatoms in this system remains highly speculative. Ciliates often form photosymbioses with green algae, although they have never acquired permanent plastids. Such interactions have not yet been observed between ciliates and diatoms, and although the possibility is intriguing, convincing evidence is lacking. We therefore recommend focussing on the most important discoveries and not on the uncertain role of diatoms in this context.

Response: We agree with the reviewer that the role of the diatoms and other microalgae inside the ciliate remains speculative at this stage. We shortened this paragraph to keep only the essential information while highlighting directions for future research to elucidate the nature of this association **line 618, page 14**, in the revised manuscript.

Lines 589-590 and 621-623 - Previously (lines 190-191), authors suggest that examined magnetotactic ciliates are unable to phagocytose prey larger than small bacteria due to the small cytostome size, contradicting themselves.

Response: This issue was also raised by Reviewer#1. Accordingly, we rephrased the section to avoid any ambiguity **line 624, page 14**, in the revised manuscript (former line 622): *“Nevertheless, additional evidence is needed to clearly define the nature of this relationship. The exact process by which diatoms and other microalgae are internalized remains unclear as direct observations are lacking. Given that diatoms and protists are similar in size, and considering the small size of the cytostome opening (0.5 μm), it is unlikely that the cytostome can expand sufficiently to engulf the entire rigid silica frustule of diatoms, as observed in other ciliates such as *Didinium nasutum*⁸⁸. ”*

Lines 623-624 - This sentence is confusing, as it seems to conflict with the prior one.

Response: The previous sentence was confusing, but after being revised according to the comment mentioned just above, it is now clear and no longer confusing .

Lines 624-625 - citation needed.

Response: The following reference has been added: Wessenberg, H. & Antipa, G. Capture and Ingestion of *Paramecium* by *Didinium nasutum*. *The Journal of Protozoology* **17**, 250–270 (1970).

Lines 602-603 - wouldn't the immediate proximity of oxygen producers be highly detrimental for all anaerobic members of the consortium?

Response: This point was also raised by the Reviewer #1, and we responded as follows: Here the ciliate appears to be living in anaerobic conditions but it might also be able to respire oxygen as most of ciliates do. Thus, it is possible that the oxygen produced by the diatom during photosynthesis is directly used by the ciliate and that prokaryotic endosymbionts are not in contact with this oxygen. It is also possible that the oxygen diffuses outside the ciliate and its time inside the ciliate is limited to avoid potential toxic effect. In this case the ciliate would benefit only from the energy produced during photosynthesis.

This part of the Discussion has been removed based on the recommendation to focus on the most important discoveries and not on the uncertain role of diatoms.

Lines 652-658 - the link between particular magnetotactic symbiosis and eukaryogenesis suggested here is unclear.

Response: In this sentence we meant that the evolution of collective magnetotaxis based on an endosymbiosis between a ciliate and biomineralizing bacteria open questions on the origin of magnetoreception in eukaryotes. As mentioned by Reviewer #1, we should have been more specific in this sentence. We rephrased it *line 659, page 14*, in the revised manuscript as follows: *“Considering the current evolutionary scenarios of eukaryogenesis and eukaryotes diversification⁹, it is appealing to link these findings to the origin of magnetoreception in eukaryotes more largely. However, although the integration of magnetotactic bacteria into a modern lineage of protists is now proven, further research is needed to determine whether such integration and further organellogenesis or lateral gene transfers could have occurred in early holobionts that gave rise to the first eukaryotes lineages able of magnetoreception.”*

Reviewer #3 (Remarks to the Author):

The manuscript by Bolzoni et al. presents the identification a new class of magnetotactic eukaryotes distinguished by an symbiotic relationship between a magnetotactic bacterium and a protist. Previous work, including groundbreaking studies from this group, had shown that protists can be magnetically sensitive in two different ways: passively through the ingestion of magnetotactic bacteria and actively through an ectosymbiotic relationship with a magnetotactic bacterium. This work presents the incredible discovery of an endosymbiotic magnetotactic bacterium. The cell biology, multi-layered endosymbiotic relationships, minerology, phylogeny, and genomics of this new group of protists is detailed through a series of carefully performed experiments. The findings are highly impactful and open up a host of new questions for microbiologists, cell biologists, and evolutionary biologists. I am thoroughly impressed by the work and cannot wait to see it published. I have only a few minor comments which can be addressed through changes to the text and figures.

Response: We thank the reviewer for their kind comments. We deeply appreciate your feedback, it's very encouraging and motivating.

1. Line 247: There is no reference to a figure but there is a closed parens.

Response: Correct. Reference to Fig. 3d and “e” has been added.

2. Line 253: There is a reference to "membrane similar to bacterial magnetosomes" in ED Figure 6d. First, I had to zoom in quite a bit to see the relevant magnetosomes. To my eyes these do not look as convincingly like the ones in the Gorby reference. Given that there is some dispute over the presence of membranes around bullet-shaped magnetosomes, I would encourage the authors to provide a more convincing image or acknowledge the possible ambiguity.

Response: It has initially been thought that bullet-shaped magnetosomes might lack a surrounding membrane (doi: 10.1073/pnas.1001290107). However, more recent studies have shown that bullet-shaped magnetites were indeed surrounded by a thin membrane

(e.g. doi:10.1111/1462-2920.13677). Although this membrane is thinner than the one described for cubo-octahedral magnetite magnetosomes in *Magnetospirillum magnetotacticum* MS-1 by Gorby et al 1988, it is now widely accepted that all types of magnetosomes are surrounded by a membrane. In **Supplementary Figure 6d**, we pointed with black arrows the bullet-shaped particles for which a membrane is visible. We believe the resolution of this figure submitted at 600 dpi allows to see the membrane of some of the bullet-shaped particles produced by the endosymbiotic strain DsD-1. These membranes have a similar contrast and thickness than those described for instance in doi:10.1111/1462-2920.13677. In the text, we modified the end of the sentence **line 249, page 6**, in the revised manuscript: “Each vesicle contains about 131 ± 26 particles ($n=13$), particles measure 143.6 ± 40.9 nm in length and 55.2 ± 8.8 nm in width ($n=629$), and are surrounded by a thin membrane similar to the membrane generally observed surrounding bullet-shaped magnetites³⁸ (**Supplementary Fig. 6d**)”. The reference from Gorby et al 1988 was replaced by doi:10.1111/1462-2920.13677.

3. Lines 339-341: I was surprised at how little attention and space was given to analyzing the magnetosome genes of the endosymbiont. This is incredibly important in thinking through the evolutionary diversity of magnetotactic bacteria and should be given a more visibility. I would encourage the authors to have a dedicated figure for these genes (extended data is fine although it could also work as a portion of a main text figure). The current extended data figure (8E) has some ambiguous descriptions in its legend. For instance, line 1425 says that each arrow represents a gene corresponding to MSR-1 operons. However, MSR-1 does not have mam genes. In the legend, “mam” should be italicized and CR-1 should be defined in the legend. I would request a table comparing the magnetosome proteins of Dsd-1 with their accession numbers to those of other magnetotactic bacteria (for instance, including the % identity to homolog in RS-1 or others). This would be a big contribution to the community.

Response: We agree with the reviewer #3 that the study of the evolution of magnetosome genes of the symbiotic magnetite producing bacteria would be very interesting. However, our assembly of DsD-1 genome enabled us to obtain a partial magnetosome gene cluster only and most of the *mam* genes usually used to study magnetotaxis evolution are missing. To get a robust phylogeny to make assumptions about magnetotaxis evolution we would need the sequence of all the magnetosome genes that are conserved in all magnetotactic bacteria including *mamABEIKLMOPQ*. We are currently attempting to improve the genome sequencing of DsD-1 to get the entire sequence of the magnetosome gene cluster and we are currently writing another manuscript specifically focusing on the evolutionary history of biomineralizing bacteria including free-living magnetotactic bacteria and symbiotic magnetosomes-producing bacteria. Please, see our response to the Reviewer #2 **page 15**, of this letter.

In the revised manuscript, we specified the relatedness of Mam proteins with their closest MTB **line 348, page 8**: “Mam proteins of DsD-1 share the highest sequence identity with their corresponding homologs in the reference strains *Solidesulfovibrio magneticus* RS-1⁵³, and *Fundidesulfovibrio magnetotacticus* FSS-1⁵⁴, with average identities of 50.6 % and 47.7 %, respectively.”. In addition a new supplementary table was provided showing these identities as proposed by the reviewer #3 (**Supplementary Table 2**)”.

We also agree that the legend of **Supplementary Figure 8** contained ambiguous descriptions. This is an error caused by copying and pasting a similar caption from a previous article. We removed the sentence “Each arrow represents a gene of a color corresponding to a specific operon in MSR-1” and replaced it by the appropriate reference in the legend of **Supplementary Figure 8e**.

4. Finally, I feel that the extrapolation from genome sequences to metabolic capability of the endosymbionts was too strong. The authors used the words "can" and "cannot" based on the presence or absence of certain genes. However, one cannot say that a certain reaction will happen because genes are there as some might be inactive. Similarly, novel pathways using genes annotated as hypothetical or as having different function may fulfill the missing pieces in a metabolic pathway. In other sentences, the authors used "may" to describe the potential activities of these organisms. This is more appropriate and should be repeated throughout.

Response: We fully agree with this comment, we revised the entire manuscript to use the conditional form to show that our genomic analyses give predictions of metabolic capability section in the revised manuscript.

5. Finally, I could not open videos 1, 2, and 5. I am not sure if it is an issue with my current setup or something is wrong with the files.

Response: These three movies are in AVI format the other movies were in MPG or MP4 formats. We checked all the videos on both a Mac and a PC and found that, depending on the software used, some videos could not be opened. We therefore converted these videos to ensure that they can be opened.

Reviewer #4 (Remarks to the Author)

Response: We thank this reviewer for their help at improving this manuscript.

Manuscript number: **NCOMMS-25-28667-T**

Title: **Magnetoreception in a freshwater ciliate arises from endosymbiosis**

Response: We would like to express our gratitude to the editor for giving us the opportunity to address the concerns raised by the reviewers in this new submission. We have carefully considered their feedback and found the overall comments and suggestions constructive. We believe we have addressed all the requested changes.

In this revised manuscript, we have provided a point-by-point response to each concern. We have included line by line where changes were made in the revised version.

Reviewers' comments/questions are written in black while our answers are written in blue.

See below.

REVIEWERS' COMMENTS

Reviewer #1 (Remarks to the Author):

I am satisfied with the revisions the authors made to the manuscript and I appreciate the time and effort they took to address all reviewers' comments. In my opinion, the revision has improved the manuscript and I am looking forward to seeing this wonderful discovery published.

Response: We thank the reviewer for their kind comments.

I have only a few last small suggestions for text edits:

Line 56: to me ,diversification' is redundant here and could be deleted

Response: The correction has been made.

Line 75-76: It is my impression that while the endosymbiotic origin of plastids and mitochondria is largely undisputed, discussions regarding the origin of nucleus persist (i.e. the autogenic models). Hence, I'd suggest to rephrase it here as such (e.g. : '... emergence of plastids, mitochondria, and potentially also the nucleus').

Response: The correction has been made, page 3, line 77: "These interactions led to the emergence of plastids, mitochondria and potentially also the nucleus."

Line 76: Consider replacing 'most' with 'many'.

Response: The correction has been made.

Line 78: mention also aerobic respiration, as the primary role of mitochondria

Response: The correction has been made.

Line 140: 'multi-partner' instead of 'multi-partners'

Response: The correction has been made.

Line 250: replace 'particles' with 'which'

Response: The correction has been made.

Line 254: consider adding 'magnetite' before 'particles' for clarity.

Response: The correction has been made.

Line 262 – 264: consider rephrasing: 'All bacteria were considered endosymbionts as they appeared to be completely enclosed ... and the extracellular space.'

Response: This sentence has been modified, page 7, line264, by: "All bacteria appeared to be fully enclosed within their host, with no detectable connection to the plasma membrane or the extracellular medium, and were therefore considered endosymbionts."

Line 272: replace 'support' with 'confirm'

Response: The correction has been made.

Line 280: replace 'genetic' with 'taxonomic'

Response: The correction has been made.

Line 299: replace 'population belongs' with 'populations belong'

Response: The correction has been made.

Line 356: As it is only 4 genomes, please add also the sizes of the two other genomes (DsD and GD-1).

Response: This sentence has been modified, page 9, line 358, by: "Genome sizes of the endosymbionts are 1.3 Mbp, 2.0 Mbp, 2.3 Mbp and 2.8 Mbp for MD-1, GD-1, DsD-1 and DcD-1, respectively (Supplementary Table 1)."

Line 363: replace 'proteins-coding' with 'protein-coding'

Response: The correction has been made.

Line 378: please italicize gene names.

Response: The correction has been made.

Line 382: what about DsD? Please add.

Response: This sentence has been modified, page 9, line 385, by: “While the DsD-1, DcD-1, GD-1, and MD-1 genomes encode 0, 3, 8, and 7 ankyrin repeat-containing proteins”.

Line 421: replace ‘account’ with ‘accounts’

Response: The correction has been made.

Line 428 – 431: sentence is unclear. Does ‘exclusively’ mean that only menaquinone and no other quinone was detected in DsD or that menaquinone was exclusively detected in DsD and in no other symbiont? Please clarify. Also, it is not obvious to me how can the quinones of one endosymbiont have such broad implications for the whole consortium?

Response: Only menaquinone and no other quinone was detected in the genome of DsD-1. The implication of the quinones of DsD-1 on the whole consortium has been added, pages 9, line 430: “The biomineralizing symbiont seems to play a central role in respiration and electron flux within the whole consortium, as only the Dsd-1 genome harbours genes encoding the synthesis of a low-potential menaquinone that is central to anaerobic respiration⁶⁹”.

Line 433: Do you mean ‘or’ instead of ‘of’?

Response: This sentence is correct, we mean “or” in the sentence page 10, line 435: “None of the two genomes encodes for cytochrome c-type or cbb3-type oxidases or other enzymes associated to inorganic terminal electron acceptors (Supplementary Table 9).”

Line 453: I suggest to slightly rephrase, e.g.: ‘A recent discovery of marine magnetic holobionts revealed a unique ability of certain microeukaryotes ...’

Response: The correction has been made.

Line 472: replace ‘with’ with ‘from’

Response: The correction has been made.

Line 473 – 474: replace with ‘...not possible as the genome of the host could not be assembled ...’

Response: The correction has been made.

Line 562-565: 'On the other hand, DcD-1 and DsD-1 could enable their host to breathe sulphate instead of oxygen.'

Please rephrase. Indeed, DcD-1 and DsD-1 can breathe sulphate, and thus so can the holobiont but not the host (= ciliate) as there is no evidence that the ciliate can access the energy obtained from sulfate reduction, unlike in ref. 81. The authors themselves proposed that the ciliate obtains energy from fermentation in the hydrogenosomes. This should be clarified.

Response: This sentence has been modified, page 12, line 564, by: "On the other hand, the host could access energy produced by its sulphate-reducing endosymbionts DcD-1 and DsD-1."

Line 653: replace 'highlights' with 'reports the discovery of' and replace 'capable to sense' with 'capable of sensing'

Response: The correction has been made.

Figure Legend:

Fig. 1b: In the text white and grey arrows are mentioned but in the image there are only black arrows. Can you check and correct if necessary?

Response: The legend of Fig. 1b has been modified by: "b, transmitted CLSM image of a magnetotactic protist showing contrasted granules in the anterior part and dark lines aligned in the anteroposterior axis (black arrows)." "grey" has been replaced by "black" while the reference of "white arrow" was unnecessary since the contrasted granules are obvious on the image and therefore do not need to be indicated on the image.

Fig. 4a: I can only see 2 arrows (=2 cells with magnetosomes), whereas 4 are mentioned in the figure legend.

Response: Panel a corresponds to the entire upper grey panel and contains three electron microscopy images: the central image shows the whole protist, while the left and right images are higher-magnification views, each displaying two endosymbionts containing magnetosomes, which are indicated by two white arrows. We have slightly modified the legend of panel a for greater clarity "(white arrows indicated in the left and right images)".

Fig. 5a,b,c,d: the symbionts from this study should be in bold in the respective trees but they are not (in my version of the figures). Please check and make sure to highlight them (in bold or colour).

Response: The correction has been made, a new Figure 5 has been submitted.

Line 1356: Replace 'under study' with 'from this study'.

Response: The correction has been made.

Fig. 6b: replace 'preys' with 'prey' in sketch

Response: The correction has been made, a new Figure 6 has been submitted.

Line 1377: replace 'cell host' with 'host cell'

Response: The correction has been made.

Line 1390 - 1392: These two sentences are essentially identical, please delete one.

Response: The correction has been made.

Reviewer #2 (Remarks to the Author):

The authors have carefully addressed our comments, as well as those of the other reviewers, and have made substantial improvements to the manuscript. All suggested changes were thoroughly considered and implemented where possible, either through additional analyses or by removing debatable statements. The main story is now much more clearly articulated and supported by additional data, while more speculative aspects have been either removed or appropriately rephrased.

Some of our comments proposing additional analyses, such as phylogenetic analysis of mam genes or more in-depth comparative analyses, were intended as general suggestions. While these were not fully pursued, we understand and agree with the authors' decision, as incorporating them would have substantially expanded the scope of the manuscript and blurred the main focus.

We have no further comments, and we would like to congratulate the Authors on this exciting and well-supported discovery in protist symbiosis and look forward to seeing the paper published.

Response: We thank the reviewer for their kind comments.

Reviewer #3 (Remarks to the Author):

The authors have answered all of my concerns. Congratulations on a fantastic manuscript.

Response: We thank the reviewer for their kind comments.

Reviewer #4 (Remarks to the Author):

Response: We thank the reviewer for their kind comments.